# Morphological differentiation of peritumoral brain zone microglia

G. Anahí Salas-Gallardo[1,2], Jonathan-Julio Lorea-Hernández[2], Ángel Abdiel Robles-Gómez[2], Claudia Castillo-Martin Del Campo[1], Fernando Peña-Ortega[2]*

1 Laboratorio de Células Neurales Troncales, CIACYT-Facultad de Medicina, Universidad Autónoma de San Luis Potosí, San Luis Potosí, San Luis Potosí, México, 2 Departamento de Neurobiología del Desarrollo y Neurofisiología, Instituto de Neurobiología, Universidad Nacional Autónoma de México, Querétaro, México

* jfpena@unam.mx

**Data Availability Statement:** All relevant data are within the paper and its Supporting Information files.

## Abstract

The Peritumoral Brain Zone (PBZ) contributes to Glioblastoma (GBM) relapse months after the resection of the original tumor, which is influenced by a variety of pathological factors. Among those, microglia are recognized as one of the main regulators of GBM progression and probably relapse. Although microglial morphology has been analyzed inside GBM and its immediate surroundings, it has not been objectively characterized throughout the PBZ. Thus, we aimed to perform a thorough characterization of microglial morphology in the PBZ and its likely differentiation not just from the tumor-associated microglia but from control tissue microglia. For this purpose, Sprague Dawley rats were intrastriatally implanted with C6 cells to induce a GBM formation. Gadolinium-based magnetic resonance imaging (MRI) was performed to locate the tumor and to define the PBZ (2 mm beyond the tumor border), thus delimitating the different regions of interest (ROIs: core tumoral zone and immediate interface; contralateral striatum as control). Brain slices were obtained and immunolabeled with the microglia marker Iba-1. Sixteen morphological parameters were measured for each cell, significative differences were found in all parameters when comparing the four ROIs. To determine if PBZ microglia could be morphologically differentiated from microglia in other ROIs, hierarchical clustering analysis was performed, revealing that microglia can be separated into four morphologically differentiated clusters, each of them mostly integrated by cells sampled in each ROI. Furthermore, a classifier based on linear discriminant analysis, including only three morphological parameters, categorized microglial cells across the studied ROIs and showed a gradual transition between them. The robustness of this classification was assessed through principal component analysis with the remaining 13 morphological parameters, corroborating the obtained results. Thus, in this study we provided objective and quantitative evidence that PBZ microglia represent a differentiable microglial morphotype that could contribute to the recurrence of GBM in this area.

## Introduction

Microglia are the resident phagocytes of the central nervous system that normally support and protect neuronal function [1–3]. Pío del Río Hortega [4], first described microglial

**Funding:** G.A.S.G. received a graduate fellowship from CONACyT (I.D. 770620). This work was supported by Dirección General de Asuntos del Personal Académico UNAM (Grant IG200521) and CONACyT (A1-S-7540). The funders had no role in study design, data collection and analysis, decision to publish, or preparation of the manuscript.

**Competing interests:** The authors have declared that no competing interests exist.

morphology; since then, it has become an outstanding feature that has been closely related to microglial functional state [5–7]. In fact, the process of microglial transformation from a surveillant phenotype to a variety of activated states is accompanied by marked morphological changes [5–7].

Several studies have revealed that microglia exhibit morphological changes during neuroinflammation and neurological diseases [2, 8–14], although those morphological phenotypes are seemingly not specific to any given condition and there is no consensus on the feature(s) that define them [13]. To deal with this problem, the quantification of precise morphometric parameters, along with clustering analysis, linear classifiers and dimension reduction algorithms, have allow researchers to classify microglia and identify microglial subtypes related to specific pathological conditions [9, 15–18].

Microglia are closely related to the evolution of glioblastoma (GBM; [19, 20]). GBM-associated microglia promote glioma cell migration [21], invasion and growth [22, 23], to the extent that microglia depletion reduces glioma volume [24] and microglia modulation has been proposed to treat these tumors [25, 26]. GBM is the most common and aggressive diffuse glioma of astrocytic lineage [27, 28]. The average life expectancy of patients with a GBM is 15 months after diagnosis [29]. Despite the advances in surgical approaches, radiotherapy and adjuvant chemotherapy, the prognosis remains poor [30]. The first therapeutic approach for GBM is usually tumor resection, which certainly improves symptoms, overall survival, and quality of life [31]. However, performing a full surgical GBM resection is nearly impossible for two main reasons. First, cancerous cells commonly migrate into the healthy parenchyma and, second, removal of large brain areas has devastating consequences [32]. Thus, recurrence is almost certain 6–9 months after diagnosis and treatment [33]. In 90% of cases, tumor recurrence emerges at the margin of the resection area [34, 35], in the peritumoral brain zone (PBZ; [36, 37]). The PBZ has been defined as radiologically normal peritumoral tissue 2 cm around the brain/tumor interface. It lacks a gadolinium-enhanced signal and has a normal appearance in T1-weighted sequences [37].

A crucial factor for tumor growth, transformation and metastasis is the tumor microenvironment (TMI), which for GBM contains non-neoplastic cells (e.g., microglia) that constitute 30 to 50% of the tumor mass [19, 36]. As principal contributors to the TMI, microglia- released factors lead to extracellular matrix degradation, which promotes glioma cell invasion and migration, thus supporting GBM progression and recurrence in the PBZ [20, 36]. Despite the potential relevance of microglia in GBM recurrence, the morphological characterization of microglia in this region is almost absent [36, 38], and its morphological differentiation from control microglia and microglia within the tumor and its surroundings has not been performed (although their molecular differentiation has been emerging; [36, 39–41]). Thus, we injected C6 glioma cells in rat brains to induce a tumor [42] in order to evaluate the morphological profile of microglia in the PBZ and compared it with that of microglia in different tumoral areas (i.e. the tumor, its interface, and the contralateral control tissue). We used clustering and dimensional reduction analysis to evaluate the likely separation of different morphological subtypes of microglia [9, 11, 15–17].

## Materials and methods

### C6 cell culture

The rat glioma C6 cell line was obtained from the American Type Culture Collection (ATTC® CCL-107™, Manassas, VA). Cells were cultured in Dulbecco´s Modified Eagle Medium (DMEM) supplemented with 10% fetal bovine serum, 100 U/mL penicillin and 100 g/mL streptomycin. Cells were grown in a humidified atmosphere with 5% $CO_2$ at 37˚C.

When the culture reached a confluence greater than 90%, it was separated from the substrate to perform cell count and viability tests with trypan blue. The minimum viability of the suspension ($1x10^5$ cell/μL) of successful cultures was set to 90%.

## Animals

All experimental procedures were approved by the Bioethics Committee of the Institute of Neurobiology at UNAM (local IACUC) and performed in accordance with the guidelines of the Official Mexican Standard for the Use and Care of Laboratory Animals (Norma Oficial Mexicana NOM-062-ZOO-1999). Male Sprague Dawley rats (230-250g; n = 5) were obtained from the breeding colony of the Institute of Neurobiology animal facility. Animals were housed in individual cages, in a temperature-controlled room (22 ± 1°C). All animals were kept under normal 12 h light-12 h dark cycle, with food and water *ad libitum*. Six animals were inoculated with the C6 cells (only one did not develop a tumor) and three animals were sham treated (see below). All animals survived and their brain were histologically processed (see below).

## Implantation surgery

Rats were anesthetized with a mixture of Ketamine/Xylazine (80 mg/kg and 10 mg/kg, respectively, i.p.) and placed in a stereotaxic frame (Stoelting Co., IL) for surgery [43]. After cleansing the skin with povidone-iodine solution, an incision was made, and a burr hole was drilled into the skull 2.5 mm lateral to the midline and 0.3 mm posterior to Bregma [44]. The injector was introduced until a depth of 5 mm from the skull was reached in a period of 2 minutes. Using a 10 μL Hamilton syringe, $5 x 10^5$ C6 cells in 5 μL Hank's Balanced Salt Solution were microinjected into the right striatum, with an injection pump, at a rate of 0.5 μL/min. The needle was kept in place for 5 minutes after the injection was completed to avoid reflux of the cell suspension through the needle path [43]. The skin was sutured, and analgesics and saline were administered intraperitoneally to prevent infection and dehydration. Animals were transferred to their housing cage after their complete recovery from anesthesia. The tumor was allowed to develop for four weeks.

## Magnetic resonance imaging

Four weeks after surgery, animals underwent a session of magnetic resonance imaging (MRI; Fig 1A; [45]). Anesthesia was induced with a 4% air/isoflurane mixture and rats were placed into the resonator with their snout fixed to a support bar. Isoflurane concentration was reduced to 2% to maintain anesthesia during image acquisition. Body temperature was preserved at 37°C using a warm water circulation system. The animal's respiration was continuously monitored with a piezoelectric sensor (40–60 breaths/minute; [45–48]). Anesthesia was stopped at the end of the session and animals were observed until complete recovery before being transported back to their place of accommodation to later be processed for histochemical studies [2, 46–49].

MRI acquisition protocols were performed with a 7 Tesla MRI scanner (Bruker Pharmascan 70/16US; 39) interfaced to a Paravision 6.0.1 console, using a 2 x 2 surface array coil [45]. A T1-weighted sequence scan (T1_FLASH_3D) lasting 18 minutes was acquired with the following parameters: repetition time (TR) = 25.63 ms, echo time (TE) = 6.99 ms, flip angle = 20°, field of view (FOV) = 36.125 x 36.125 x 19.2 (mm), matrix = 290 x 290, number of slices = 64. Then, gadolinium contrast medium was administered through the jugular vein (0.5 mmol/kg; Fig 1A). A new T1-weighted sequence was performed with the same parameters, and volumetric images were obtained for further study.

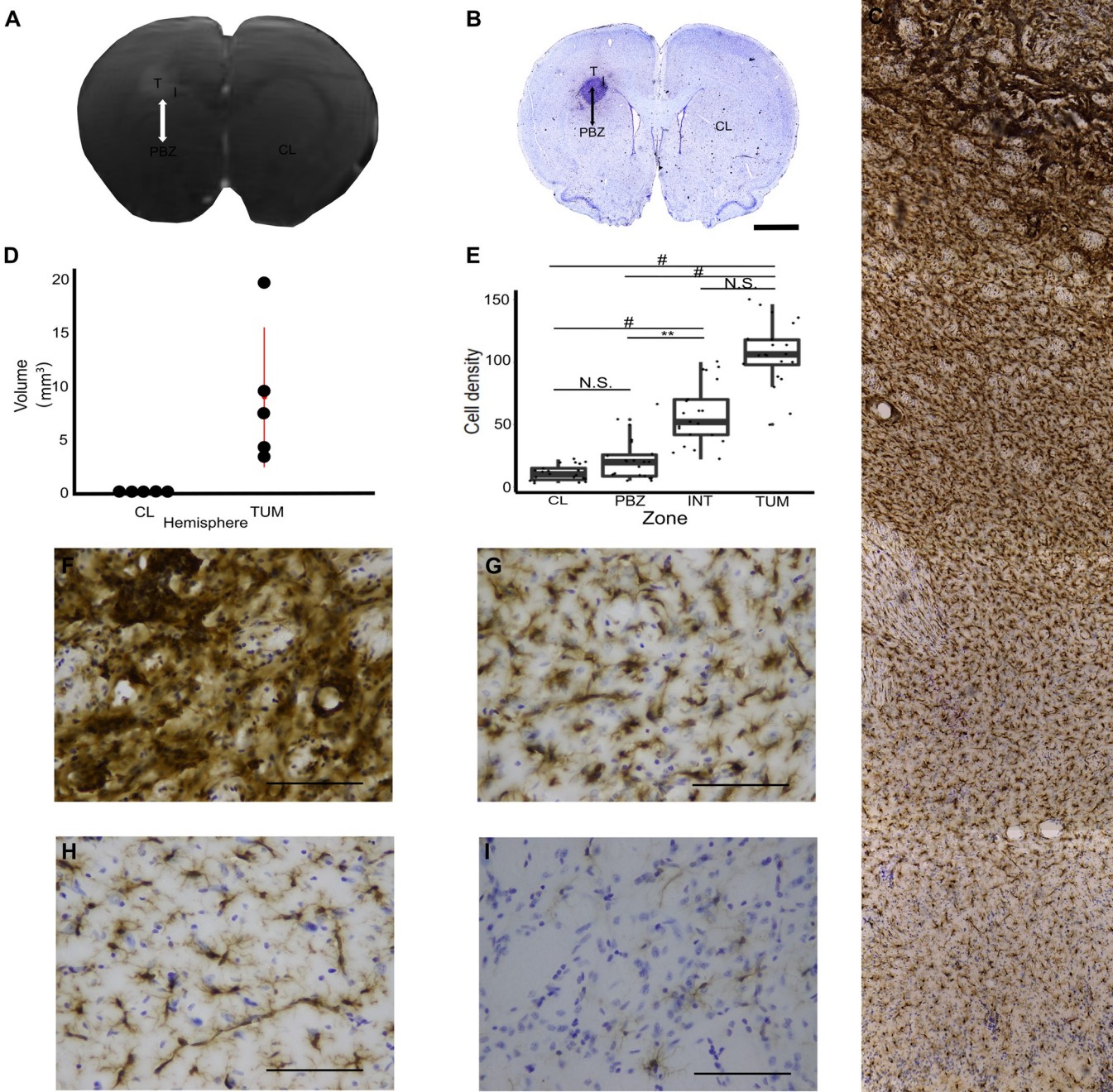

**Fig 1. Magnetic Resonance Imaging (MRI)-histological correspondence and microglial cell density in the tumor, its surroundings, the peritumoral brain zone (PBZ) and control tissue peritumoral brain zone (PBZ) localization. (A)** Gadolinium-enhanced MRI sample of a coronal brain gadolinium-enhance section containing the tumor. The PBZ is delimited by a double-headed arrow. **(B)** Mosaic reconstruction of the Iba-1-immunostained/Nissl counterstained slice matching the MRI sample. Scale bar represents 2 mm. Both images exhibit the four regions of interest (ROI; T = tumor, I = interface, PBZ, CL = contralateral hemisphere). **(C)** Iba-1-immunostained/Nissl counterstained tumor and its surroundings (10X objective). **(D)** Tumor volume for each animal compared to its contralateral hemisphere. **(E)** Quantification of the microglia density in an area of 25.96 x $10^3$ $\mu m^2$ for each ROI. *P<0.05, **P<0.01, #P<0.001. Representative photomicrographs (40X objective; Scale bar represents 50 $\mu$m) of Iba-1-immunostained/Nissl counterstained **(F)** tumor, **(G)** interface, **(H)** PBZ and **(I)** CL areas are also shown.

## Immunostaining

Up to 2 h after the MRI session, rats were anesthetized with a mixture of Ketamine/Xylazine (80 mg/kg and 10 mg/kg, respectively, i.p.) and intracardially perfused with 0.01 M phosphate-buffered saline (PBS), followed by with 4% paraformaldehyde (PFA) in PBS [2, 43, 48, 49]. The brain was removed and incubated in 4% PFA overnight. Then, it was transferred to a 30% sucrose solution until saturation. Saturation was achieved when the brain sank to the bottom of the 30% sucrose solution [43, 49]. Coronal sections (40μm thick) were obtained with a cryostat (-18˚C). Two slices per animal (10 slices from 5 animals), around 280 μm apart and including the tumor, were selected for immunohistochemistry (Fig 1; [2, 43, 48]).

Free floating sections were incubated for 30 minutes in 10% methanol and 3% hydrogen peroxide solution to inhibit endogenous peroxidase [48]. After washing with PBS, non-specific binding sites were saturated with 2% bovine serum diluted in PBS containing 0.05% tween 20, 0.1% Triton X-100 and 50 nM glycine for 30 minutes. Microglia labelling was carried out with the polyclonal anti-Iba-1 primary antibody (rabbit anti-Iba1, 1:300, #019–19741, Wako, Osaka, Japan; Fig 1B and 1C; [2, 48]), which was applied to the slices overnight at 4˚C in 0.05% tween 20, 2% bovine serum, 0.1% Triton X-100 and 10 nM glycine in PBS. Then, slices were washed and incubated with biotinylated secondary antibody specific for rabbit immunoglobulin heavy chains (Goat Anti-Rabbit IgG Antibody (H+L), Biotinylated (BA-1000-1.5), Vector laboratories) at room temperature for 1.5 h. The biotin-avidin complex amplification system (ABC, 1:250; Thermo Fisher Scientific) was used to detect biotinylated secondary antibody [48]. Peroxidase activity was revealed with a solution of 0.05% diaminobenzidine and 0.03% hydrogen peroxide in PBS for approximately 10 min [48]. After washing with PBS, the slices were mounted on gelatinized slides and air-dried. Counterstaining was performed with cresyl violet (0.1%; Sigma, USA; 2,42–45) to identify cell bodies (Fig 1B; [43, 49–51]). At the end of this procedure, the slides were dehydrated in ethanol, xylene, and a coverslip with DPX mounting medium was placed [2, 42].

## MRI processing

In the MRI images, the skull was stripped, and noise removed using random matrix theory with the DenoiseImage command available in the ANTs (Advanced Neural Tools) Software. The bias of the field inhomogeneities was corrected using the ANTs Software with the N4 algorithm (N4BiasFieldCorrection version 2.2.0, available in ANTS, [52]). Tumor volume ($mm^3$) was measure for each animal using the ITK-SNAP Software (version 3.8.0; Fig 1D; [53]). The tumors were discrete and well-demarcated, likely due to the reduced amount of transfected cells and the host strain used [54], as has been found by others [42, 54–57].

## MRI-histology correspondence

Lemée et al. (2015) defined the PBZ as a radiologically normal peritumoral area, located within a 2 cm distance from the brain/tumor interface. Considering the proportion between human and rat brain mass (0.00119; [58, 59]) we considered the PBZ in the rat as a radiologically normal peritumoral area, located within a 2 mm distance from the brain/tumor interface. This area exhibits a normal aspect on T1-weighted sequences and gadolinium enhancement is absent (Fig 1A). Thus, to use the same criteria we matched the immunostained slices to their correspondent MRI slice before defining the regions of interest (ROIs) for microglia characterization (Fig 1A and 1B). For this purpose, each immunostained slices was fully reconstructed using 4x micrographs with Fiji software. This image was matched to a similar image in the MRI Waxholm Space atlas of the Sprague Dawley rat brain [60] which was then matched to a specific MRI volume using the ANTs Software [61]. Thus, the image from the MRI was

selected according to the immunostained slice and both images were fully matched in the Register Software https://www.mcgill.ca/bic/software/visualization/register). Then, MRI image was binarized and 2 mm from the tumor border were measured in ITK-SNAP software (version 3.8.0, 53). The PBZ and other tumor-related areas were delineated as the analogous ones defined in humans (Fig 1A and 1B; [62, 63]) as follows: the tumor zone was defined as a gadolinium-positive and high cell density region (Fig 1A; [37]). The interface zone was the gadolinium-free region immediately surrounding (around 170 μm) the tumor [37], whereas the contralateral hemisphere, in an area equivalent to the PBZ, was considered the control region (Fig 1A and 1B).

## Image acquisition

To obtain a reconstruction of each slice, mosaics were obtained in a Nikon Eclipse Ci microscope at low magnification (Nikon 4x CFI Plan Achromat Microscope Objective). To quantify the microglia density in a 40x photomicrographs ($25.96 \times 10^3$ $\mu m^2$), seven slices (from four animals) were sampled to obtain three photomicrographs for each ROI (Nikon 40x CFI Plan Achromat Microscope Objective; Fig 1F–1I). To study microglial morphology, a set of 10 image stacks (1 μm thick/10 μm depth) were acquired from the 5 animals at the highest magnification (Nikon 100x Oil CFI Plan Achromat Microscope Objective; Fig 2A). The stack was combined to obtain a high-quality cropped image (pixel size = 0.081 μm; Fig 2A).

## Image processing and measurement of morphological parameters

Eight cells per area (tumoral, interface, peritumoral and contralateral), imaged at the highest magnification, were selected per slice. For this purpose, the 40x objective was randomly directed to each of the zones. All entire and nonoverlapping cells within the randomly selected area were identified, imaged at 100x and analyzed. When the area imaged with the 40x objective included more than the required eight cells, a random number generator was used to select the ones to be incorporated in the sample. Since two slices per animal were sampled and five animals were studied, 80 cells of each zone were imaged, with a grand total of 320 cells from all ROIs included in the main analysis (Fig 4).

Two additional validation cell sets were sampled. One set of PBZ microglia included 200 cells from the 10 slices (20 cells per slice; Fig 7). The other set included 36 control microglia (obtained from the contralateral region) from the five animals (7–8 cells per animal). Ten microglial cells were sampled from a brain with no tumor (Non-tumor) and 30 microglial cells were sampled from three animals (10 cells each) that underwent sham surgery (i.e. 5 μL Hank's Balanced Salt Solution were microinjected into the right striatum). These cells were morphologically analyzed and compared to the contralateral microglia for all the parameters described in the next paragraph. Non-significant and consistent differences were found between the Non-tumor, sham and contralateral microglia in most of the 16 measured parameters (S3 Fig) pointing out a strong similarity between these tissues and discarding any influence from tumoral cells or surgery in the contralateral hemisphere. Moreover, these characteristics coincide with those reported for control microglia in other studies [9, 64–70].

The high magnification micrographs [71] were analyzed using Fiji Software (https://imagej.net/software/fiji/) as described by Young & Morrison (2018). Briefly, the blue component of the images stack was obtained and bandpass filtered (up to 3 and down to 40 pixels) before being converted to a grayscale (Fig 2B). An Unsharp Mask filter was applied and then despeckled to remove salt and pepper noise. Then, images were binarized and each cell was isolated and reconstructed manually always confirming with the original micrograph (Fig 2C). The binarized cell was outlined (Fig 2D) and converted to a skeleton using the FIJI Skeleton

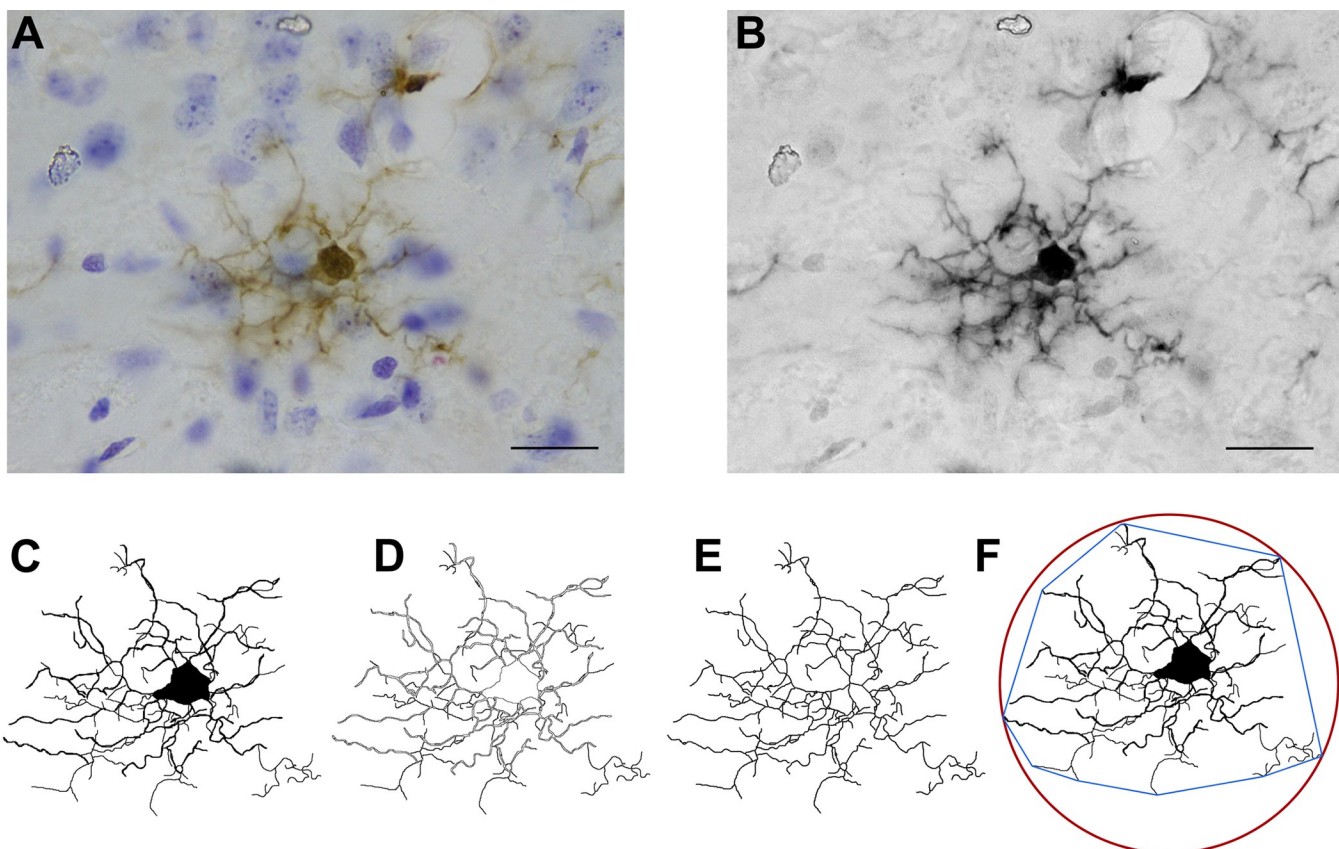

**Fig 2. Image processing for microglial morphological characterization.** **(A)** Representative high magnification (100x objective; Scale bar represents 10μm) cropped image (10 images 1 μm thick/10 μm depth) of a microglial cell immunolabeled for Iba-1 (along with Nissl counterstaining). **(B)** Grayscale and filtered image. **(C)** Binarized cell. **(D)** Outlined cell. **(E)** Skeletonized cell. **(F)** Binarized cell with its Convex Hull (blue) and bounding circle (red).

plug in (Fig 2E). All three images were used to measure the morphological parameters described next ([9]; Fig 3). FracLac and Skeleton plug-ins were used to obtain the following sixteen parameters:

1.- Number of Branches (NOB). Calculated using Analyze Skeleton (2D/3D; [59]). 2.- Fractal dimension (FD), which quantified microglia shape complexity by measuring their contour defined by the endpoints and process lengths [72]. Greater complexity of the pattern is reflected in a higher FD. 3.- Lacunarity (LAC) measures heterogeneity or translational and rotational invariance in a shape. High LAC levels are associated with shape heterogeneity (i.e., the image presents many different size gaps or lacunas), while low LAC value is associated with homogeneity. LAC was measured using the FracLac plugin as a coefficient of variation expressed in pixel density per box as a function of box size [9]. 4.- Cell area (CA) was measured as the total number of pixels in the filled microglia. 5.- Convex hull area (CHA) is the number of pixels in the convex hull (the smallest convex polygon that contains the whole cell shape; the sum of its interior angles is less than 180˚). 6.- Density (DEN) was obtained by dividing CA by its CHA. 7.- Cell perimeter (CP) represents the total pixels present in the outline cell shape. 8.- Convex hull span ratio (CHSR) was calculated as the ratio of the major over the minor axes of the Convex hull. 9.- Maximum span across the convex hull (MSACH) is the maximum distance between two points across the convex hull. 10.- Convex hull perimeter (CHP) is the single outline of the convex hull. 11.- Roughness (R) was obtained by dividing the cell perimeter

by its CHP. 12.- Cell circularity (CC) was calculated with the following formula: $(4\pi \times \text{Cell area})/(\text{Cell perimeter})^2$. The circularity value of a circle is 1 [9]. 13.- Convex hull circularity (CHC) was calculated as $(4\pi \times \text{Convex hull area})/(\text{Convex hull perimeter})^2$. 14.- Maximum/minimum convex hull ratio (TRMM) is the ratio between the largest to the smallest radius from the center of mass of the convex hull to an exterior point [9]. 15.- The mean radius (MR) is the mean length from the center of mass of the convex hull to an exterior point. 16.- Diameter of the bounding circle (DOB) is the diameter of the smallest circle that contains the convex hull (Table 1).

## Hierarchical clustering analysis

All 16 morphological parameters were used for hierarchical clustering analysis (HCA), in RStudio Software (Version 4.0.3), to classify microglial types and relate them to the ROIs from where they were sampled. To normalize the measurements, z-scores were obtained for all the parameters [15, 73]. Normalized values were used to measure the Manhattan distance between all cells [74]). Then, a dendrogram was obtained using the Ward´s linkage algorithm [75] and segmented into four branches to cluster the cells from the four sampled ROIs (Fig 4). The percentage of cells belonging to each ROI was measured in each branch and the ROI mostly represented in a branch defined the cluster identity (Fig 4). Then, cells not belonging to any given cluster were eliminated from the next analyses (77 cells; 31.69%).

## Identification of morphological parameters that better separate the defined clusters

To identify the most useful morphological parameters for differentiating microglia of the four previously defined ROIs, we measured the multimodality index (MMI) for each parameter as follows: MMI = $[M3^2 + 1] / [M4 + 3 (n - 1)^2 / (n - 2) (n - 3)]$, where M3 is skewness, M4 is kurtosis, and n is the sample size [76]. The parameters that exhibited a MMI greater than 0.55, likely presenting multimodal distributions, were CA, DEN, CP, CHSR, MSACH, CHA, CHP, CC, MR and DOB (Table 2). These parameters were subjected to a Pearson correlation analysis (Fig 4C) with the function *cor* from the STATS library in the RStudio software.

## Linear discriminant analysis

Linear Discriminant Analysis (LDA) was used as a statistical pattern classification method [77] using the *lda* function embedded in the MASS package for RStudio software. The following equation describes the linear discriminant functions:

$$Y = A_1X_1 + A_2X_2 + \ldots + A_nX_n$$

where $A_n$ is the coefficient of individual morphometric parameters and $X_n$ is each morphometric parameter. Discriminant scores were the Y values of the linear discriminant functions. The number of discriminant functions generated is always g– 1, where g is the number of groups being discriminated (four in our study; [17]). The standardized coefficients reflect the net contribution of each variable to the discriminant function [9]. Three parameters were included in this analysis: CA, DEN, and CC. The reason for including CC and DEN was because both parameters exhibited the highest MMI values (0.74 and 0.70 respectively). The combination of both parameters with CA, which exhibit the second lowest redundancy (lowest correlation) with both parameters (the lowest being CHSR), rendered the more reliable classification. All the cells grouped in the four ROIs with the HCA were included in this analysis (243 microglial cells).

## Principal component analysis

Principal component analysis (PCA) was performed in the RStudio Software to further evaluate morphological clustering of microglia using the 13 remaining measured parameters (excluding those used for LDA): NOB, FD, LAC, CP, CHSR, MSACH, CHA, CHP, R, CHC, TRMM, MR and DOB. To select the Principal Components (PCs) that represented the systematic sources of variation in our data and discard PCs that only reflect random noise, a permutation-based test was employed [78–80]. *PCAtest* (https://github.com/arleyc/PCAtest) was used to evaluate the significance of each PC and of the variable loading for the significant axis [80]. This function applies a permutation-based test and builds a null distribution to be compared for each parameter [80]. The cells grouped in the four ROIs identified with the HCA were included in this analysis (243 microglial cells).

## Statistical analysis

A Kolmogorov-Smirnov test was performed to evaluate normality of the distributions. A Kruskal-Wallis test, followed by a Dunn´s test, was mostly used to compare differences between groups. The results from the LDA were statistically tested with a Wilks's lambda and chi-squared tests. For the *PCA test* normality was evaluated by a Shapiro Wilk test. An analysis of variance (ANOVA) followed by a t-test were performed for comparisons. Differences were considerate significant with a P < 0.05. All analyses were performed using the RStudio Software Version 4.0.3.

## Results

### Microglial cell density varies between the tumor and its surroundings, the PBZ and control tissue

Before proceeding with the morphological single-cell characterization of microglia in the four ROIs—t tumoral (TUM), interface (INT), peritumoral brain zone (PBZ) and contralateral (CL)—we analyzed the microglial density in 25.96 x $10^3$ μm$^2$ of each ROI (Fig 1E–1I). We found a significant microglial density gradient ($\chi^2$(3, $N$ = 21) = 64.81 P = 5.50 x $10^{-14}$) with maximal density in the TUM zone and minimal density in the control area (CL; Fig 1E). Pairwise comparison (Dunn´s test with Bonferroni correction) indicated that cell density in the CL zone (11.09 ± 6.19 cells) was significantly lower than the density found in the INT (58.00 ± 24.32 cells, P = 1.05 x $10^{-5}$) and the TUM (106.95 ± 25.98 cells, P = 1.11 x $10^{-12}$) areas but not different from the density found in the PBZ (22.62 ± 17.12 cells; P = 0.67). The microglial density in the PBZ was significantly lower than the one found in the INT (P = 8.56 x $10^{-3}$) and TUM (P = 4.82 x $10^{-8}$) areas. Finally, no significant differences were found between the cell densities in the TUM and INT areas (P = 0.60; Fig 1E).

### Differential microglial morphology in the tumor, its surroundings, the PBZ and control tissue

To morphologically characterize immunolabeled microglia (Fig 1C), 16 parameters were measured for each cell (Table 1; S1 Fig; also see Materials and Methods). These morphological parameters were statistically compared between ROIs and most comparison rendered significant differences (P<0.05; Fig 3; S1 Fig). In fact, we found that only three parameters (CA, CHSR and CHC) were similar (P< 0.05) between the TUM and INT zones (Fig 4; S1 Fig). Moreover, only three parameters (LAC, DEN and TRMM) were similar (P< 0.05) between the CL zone and the PBZ (Fig 3; S1 Fig) and only FD was similar (P< 0.05) between the INT zone and PBZ (Fig 3; S1 Fig). As mentioned, the rest of the comparisons rendered significant

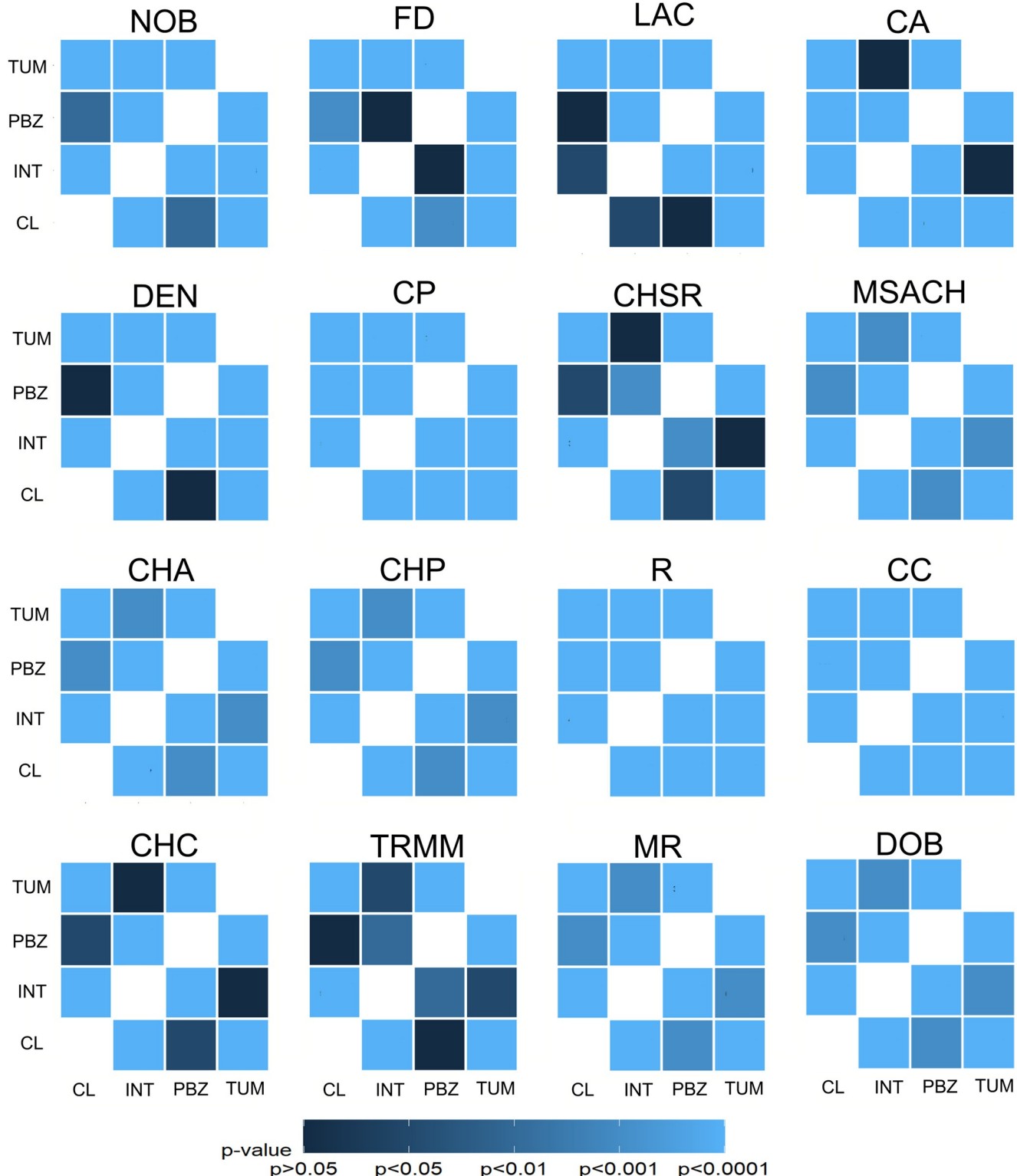

**Fig 3. Statistical comparisons among the sixteen morphological parameters measured in microglia sampled from the four ROIs.** Significance matrices were built for the following parameters: Number of branches (NOB); Fractal dimension (FD); Lacunarity (LAC); Cell area (CA); Convex hull area (CHA); Density (DEN); Cell perimeter (CP); Convex hull span ratio (CHSR); Maximum span across the convex hull (MSACH); Convex hull Perimeter (CHP); Roughness (R); Cell circularity (CC); Convex hull circularity (CHC); Maximum/minimum convex hull radius ratio (TRMM); Mean radius (MR); Diameter of the bounding circle (DOB) comparing the four regions of interest (T = tumor, I = interface, PBZ, CL = contralateral hemisphere). P-values are provided and represented in a blue scale (n = 80 cells/group).

**Table 1. Mean and standard deviation of the 16 morphological parameters of microglia sampled in the four regions of interest along with their statistical comparison.**

| Parameter | Contralateral (n = 80) | Peritumoral (n = 80) | Interface (n = 80) | Tumoral (n = 80) | Kruskal- Wallis (df) |
|---|---|---|---|---|---|
| NOB | 246.46 ± 144.04 | 137.14 ± 69.06 | 46.95 ± 33.12 | 10.99 ±7.23 | $\chi^2(3)$ = 255.66. $p< 2.2e\text{-}16$ |
| FD | 1.48 ± 0.06 | 1.42 ± 0.06 | 1.38 ± 0.07 | 1.22± 0.06 | $\chi^2(3)$ = 209.63. $p< 2.2e\text{-}16$ |
| LAC | 1.10 ± 0.24 | 1.23 ± 0.27 | 0.96 ± 0.22 | 0.61 ± 0.22 | $\chi^2(3)$ = 155.97, $p< 2.2e\text{-}16$ |
| CA($\mu m^2$) | 214.44 ± 55.86 | 137.86 ± 50.19 | 71.53 ± 42.01 | 60.06 ± 41.94 | $\chi^2(3)$ = 202.69, $p < 2.2e\text{-}16$ |
| DEN | 0.11 ± 0.03 | 0.14 ± 0.04 | 0.29 ± 0.09 | 0.56 ± 0.18 | $\chi^2(3)$ = 254.71, $p < 2.2e\text{-}16$ |
| CP($\mu m$) | 1477.42 ± 420.66 | 798.17 ± 312.40 | 257.11 ± 147.69 | 77.70 ± 34.12 | $\chi^2(3)$ = 278.41. $p< 2.2e\text{-}16$ |
| CHSR | 1.41 ± 0.25 | 1.58 ± 0.34 | 2.04 ± 0.7 | 2.25 ± 0.83 | $\chi^2(3)$ = 88.259. $p< 2.2e\text{-}16$ |
| MSACH($\mu m$) | 65.39 ± 12.48 | 49.13 ± 10.90 | 26.02 ± 6.56 | 17.82 ± 5.97 | $\chi^2(3)$ = 259.73. $p< 2.2e\text{-}16$ |
| CHA ($\mu m^2$) | 2097.07 ± 764.75 | 1099.68 ± 472.48 | 269.33 ± 180.32 | 110.93 ± 70.15 | $\chi^2(3)$ = 265.05. $p< 2.2e\text{-}16$ |
| CHP($\mu m$) | 174.78 ± 31.06 | 127.76 ± 27.65 | 65.07 ±17.97 | 43.11 ± 13.74 | $\chi^2(3)$ = 264.57. $p< 2.2e\text{-}16$ |
| R | 8.48 ± 1.95 | 6.08 ± 1.55 | 3.74 ± 1.21 | 1.76 ± 0.36 | $\chi^2(3)$ = 264.59. $p< 2.2e\text{-}16$ |
| CC | 0.0013 ± 0.0005 | 0.0037 ± 0.003 | 0.02 ± 0.01 | 0.15 ± 0.09 | $\chi^2(3)$ = 280.16. $p< 2.2e\text{-}16$ |
| CHC | 0.84 ± 0.05 | 0.80 ± 0.06 | 0.72± 0.10 | 0.69 ± 0.12 | $\chi^2(3)$ = 98.144. $p< 2.2e\text{-}16$ |
| TRMM | 1.58 ± 0.26 | 1.74 ± 0.36 | 2.05 ± 0.63 | 2.39 ± 0.87 | $\chi^2(3)$ = 86.583.. $p< 2.2e\text{-}16$ |
| MR($\mu m$) | 29.03 ± 5.24 | 21.44 ± 4.58 | 11.24 ± 2.98 | 7.72 ± 2.56 | $\chi^2(3)$ = 261.47. $p< 2.2e\text{-}16$ |
| DOB($\mu m$) | 65.88 ± 12.33 | 49.43 ± 10.83 | 26.20 ± 6.62 | 17.90 ± 5.98 | $\chi^2(3)$ = 260.45. $p< 2.2e\text{-}16$ |

Number of branches (NOB); Fractal dimension (FD); Lacunarity (LAC); Cell area (CA); Convex hull area (CHA); Density (DEN); Cell perimeter (CP); Convex hull span ratio (CHSR); Maximum span across the convex hull (MSACH); Convex hull perimeter (CHP); Roughness (R); Cell circularity (CC); Convex hull circularity (CHC); Maximum/minimum convex hull radius ratio (TRMM); Mean radius (MR); Diameter of the bounding circle (DOB).

differences between ROIs (Fig 3; S1 Fig). Similarly, as a corroboration, we performed a Non-Parametric MANOVA (NPMANOVA) using the *adonis* function from *vegan* package in RStudio Software [36]. All 16 measured parameters were evaluated for the 4 ROIs and significative differences were found (F(3) = 262.99, p = 0.001). A wrapper function for multilevel pairwise comparison using the *pairwise.adonis* function with Bonferroni correction was used as post-hoc test. The following results were obtained: CL vs PBZ (F(1) = 97.339504, p = 0.01), CL vs INT (F(1) = 470.320998, p = 0.01), CL vs TUM (F(1) = 723.741213, p = 0.01), PBZ vs INT (F(1) = 194.567536, p = 0.01), PBZ vs TUM (F(1) = 426.054392, p = 0.01), INT vs TUM (F(1) = 81.390667, p = 0.01) (S2 Fig).

## Hierarchical clustering segregates microglia from the tumor, its surroundings, the PBZ and control tissue

HCA using the Ward´s method with the Manhattan-distance, including the 16 parameters measured in the 320 cells sampled from the four ROIs, render a dendrogram (Fig 4A) that clearly exhibits four branches, as previously reported [9, 15, 17]. To test if those four branches are mostly integrated by cells sampled from each of the four ROIs (TUM, CL, INT and PBZ), we calculated the proportion of cells in each branch belonging to these ROIs and found that this, indeed, was the case (Fig 4B). One branch was mostly constituted by cells from the tumor (TUM; 83.13%), another branch was mostly constituted by cells from the CL region (71.72%), a third branch was mostly made up of cells from the INT zone (78.57%) and, as aimed to reveal in this study, one branch was mostly formed by cells from the PBZ (70.59%; Fig 4B). Thus, HCA revealed that PBZ microglia can be morphologically differentiated from those of the tumor, its immediately surroundings (INT) and from control tissue (CL; Fig 4B). However, all these distinctive morphological clusters included a minority of cells from different ROIs (from

**Table 2. Multimodality index of the 16 morphological parameters of microglia sampled in the four regions of interest and the coefficients for the three morphological parameters (\*) used in linear discriminant analysis.**

| Parameter | MMI | LD1 | LD2 | LD3 |
|---|---|---|---|---|
| CA* | 0.5676 | -1.4782 | 1.2634 | -0.6258 |
| DEN* | 0.696 | 2.9648 | 0.5499 | -2.3319 |
| CC* | 0.7438 | -1.1586 | 0.6723 | 2.2051 |
| NOB | 0.4813 | | | |
| FD | 0.4684 | | | |
| LAC | 0.3971 | | | |
| CP* | 0.6435 | | | |
| CHSR* | 0.5976 | | | |
| MSACH* | 0.5602 | | | |
| CHA* | 0.583 | | | |
| CHP* | 0.5765 | | | |
| R | 0.4692 | | | |
| CHC | 0.509 | | | |
| TRMM | 0.5427 | | | |
| MR* | 0.5744 | | | |
| DOB* | 0.5592 | | | |
| | | *0.909* | *0.0866* | *0.0045* |

Linear discriminant 1, 2 and 3 (LD1; LD2 and LD3; respectively) Number of Branches (NOB); Fractal dimension (FD); Lacunarity (LAC); Cell area (CA); Convex hull area (CHA); Density (DEN); Cell perimeter (CP); Convex hull span ratio (CHSR); Maximum span across the convex hull (MSACH); Convex hull perimeter (CHP); Roughness (R); Cell circularity (CC); Convex hull circularity (CHC); Maximum/minimum convex hull radius ratio (TRMM); Mean radius (MR); Diameter of the bounding circle (DOB). The values in italics represent the proportion of separation (i.e. total variance) provided be each linear discriminant.

5.71 to 28.28%). These minority cells (77 cells) were excluded from the rest of the analyses. Thus, the next analyses were performed with 243 cells that constitute most of their respective clusters.

## Few morphological parameters (CA, DEN and CC) can differentiate microglia from the tumor, its surroundings, the PBZ and control tissue using LDA

Once we stablished that all 16 parameters can be used to separate PBZ microglia and microglia from the other ROIs using HCA, we tested if a fewer set of parameters can achieve the same aim using a classifier to predict the allocation of a microglial cell within its correspondent cluster (Fig 5). To select the appropriate parameters, the multimodality index was calculated (MMI, See Materials and Methods), as reported previously [9, 15], aiming to identify parameters with MMI>0.55 (i.e. multimodal distributions; [9, 15]; Table 2). Ten out of the 16 morphological parameters presented an MMI>0.55 (Table 2); but CC and DEN exhibited the highest MMI values (0.74 and 0.70, respectively) and thus were selected for the classifier. In addition, a correlation analysis was performed (Fig 4C) to select other parameter with low correlation whit CC and DEN to further feed the classifier. We found that CHSR exhibited the lowest correlations with CC and DEN (Fig 4C) but LDA analysis using these three parameters render a poor performance (83.33% of correctly classified cells; i.e., 60 out of 72 cells; see below). Alternatively, combining CC and DEN with CA, the parameter with the second lowest correlation values with CC and DEN, rendered a LDA analysis that significantly separated the distinct microglial populations, with TUM and CL cells located at opposite ends of the LD1-2 space, INT cells closer to TUM cell and PBZ closer to CL cells (Fig 5). Linear discriminant

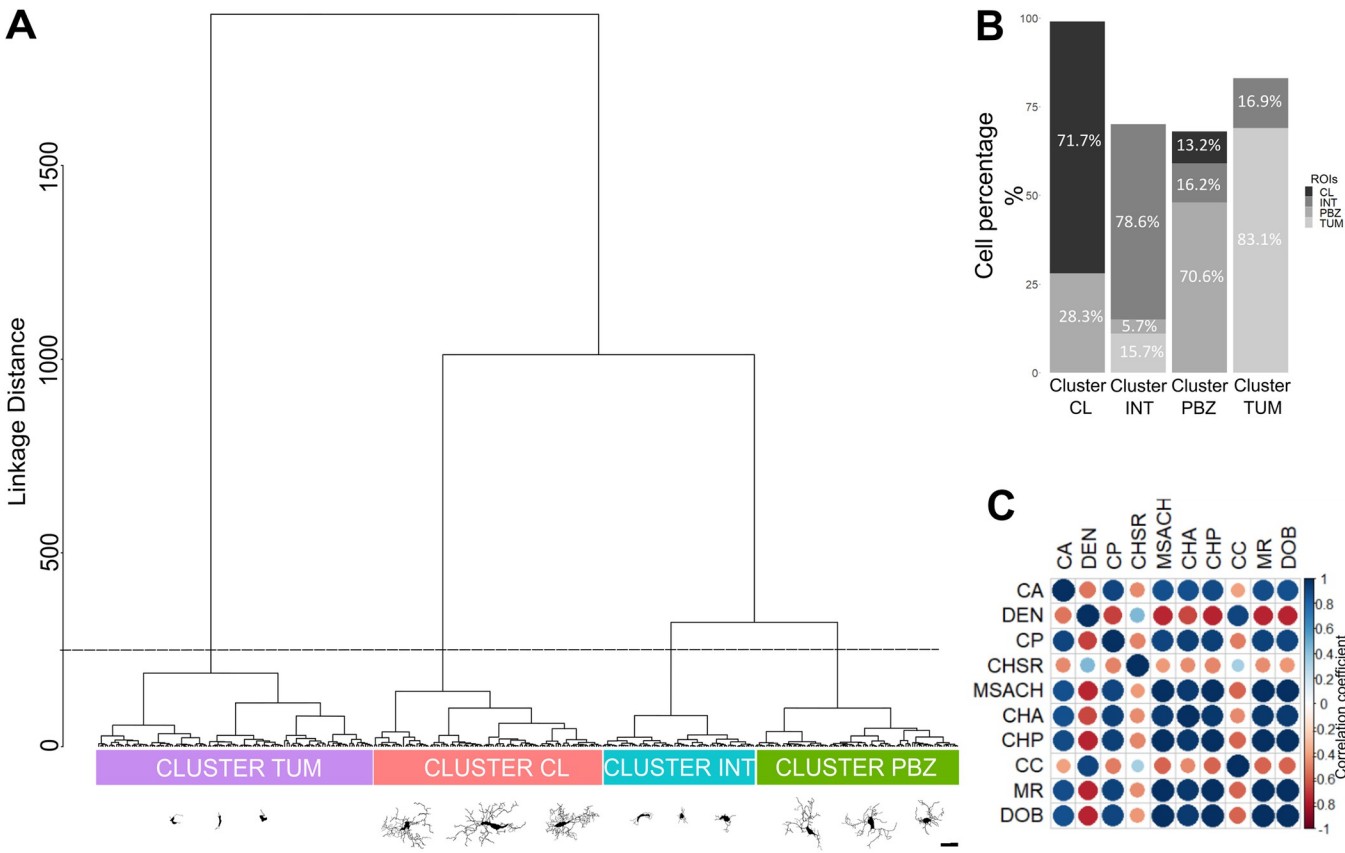

**Fig 4. Hierarchical Cluster Analysis (HCA) differentiates four clusters, mostly integrated by cells from the four ROIs. (A)** Dendrogram obtained with HCA, using sixteen morphological parameters of 320 microglial cells (represented in the X axis; three representative examples of each morphological cluster are presented; Scale bar = 10 μm), evaluating their Manhattan Distance (Y axis). The dashed line indicates the cut off level to obtain four clusters. **(B)** Proportional composition of the four branches (clusters) of the dendrogram. The grayscale represents the cells´ original ROI. Each branch (cluster) was identified according to the ROI mostly represented. **(C)** Correlation matrix between the morphological parameters with multimodality indexes greater than 0.55 (ten parameters): Cell Area (CA); Convex Hull Area (CHA); Density (DEN); Cell perimeter (CP); Convex Hull Span Ratio (CHSR); Maximum span across the Convex Hull (MSACH); Convex Hull Perimeter (CHP); Cell circularity (CC); Mean radius (MR); Diameter of the Bounding Circle (DOB). Circles size is proportional to the absolute value of the correlation coefficient. The actual correlation coefficient is represented in a color scale.

function, fed with CC and DEN with CA parameters, was trained with 171 cells (~70%) and tested with 72 cells (~30%). With these parameters the first lineal discriminator (LD1) explained 90.90% of total variance, while the second lineal discriminator (LD2) explained 8.66% of total variance (Table 2). Moreover, the classifications achieved with this method and parameters were significant (Wilks's lambda = 0.02; chi-squared = 257.36; df = 9; P< 2.20 x $10^{-16}$). The linear discriminator coefficients (Table 2) reveal that DEN is the strongest predictor for LD1 while CA is the strongest predictor for LD2 (Table 2). After training the classifier with 171 cells (~70%) and testing it with 72 cells (~30%), the classifier correctly classified 94.44% of the cells (i.e., 68 out of 72 cells; Fig 5A). We further challenged the classifier with an independent set of 36 control cells (Fig 5C), with an independent set of 200 PBZ cells (Fig 5D), or with their combination (Fig 5B). The results of the combined sample of independent PBZ and control cells (Fig 5B) indicated that 62.7% of these cells were correctly classified and a significative difference between clusters was found (Wilks's lambda = 0.24352; chi-squared = 327.72; df = 6; p-value < 2.2e-16). However, 14 of the 236 cells were wrongly classified as belonging to the INT zone (no cell was classified as tumoral). The analysis was repeated separating PBZ and CL cells (Fig 5C and 5D). When testing the CL microglial cells, 97.22%

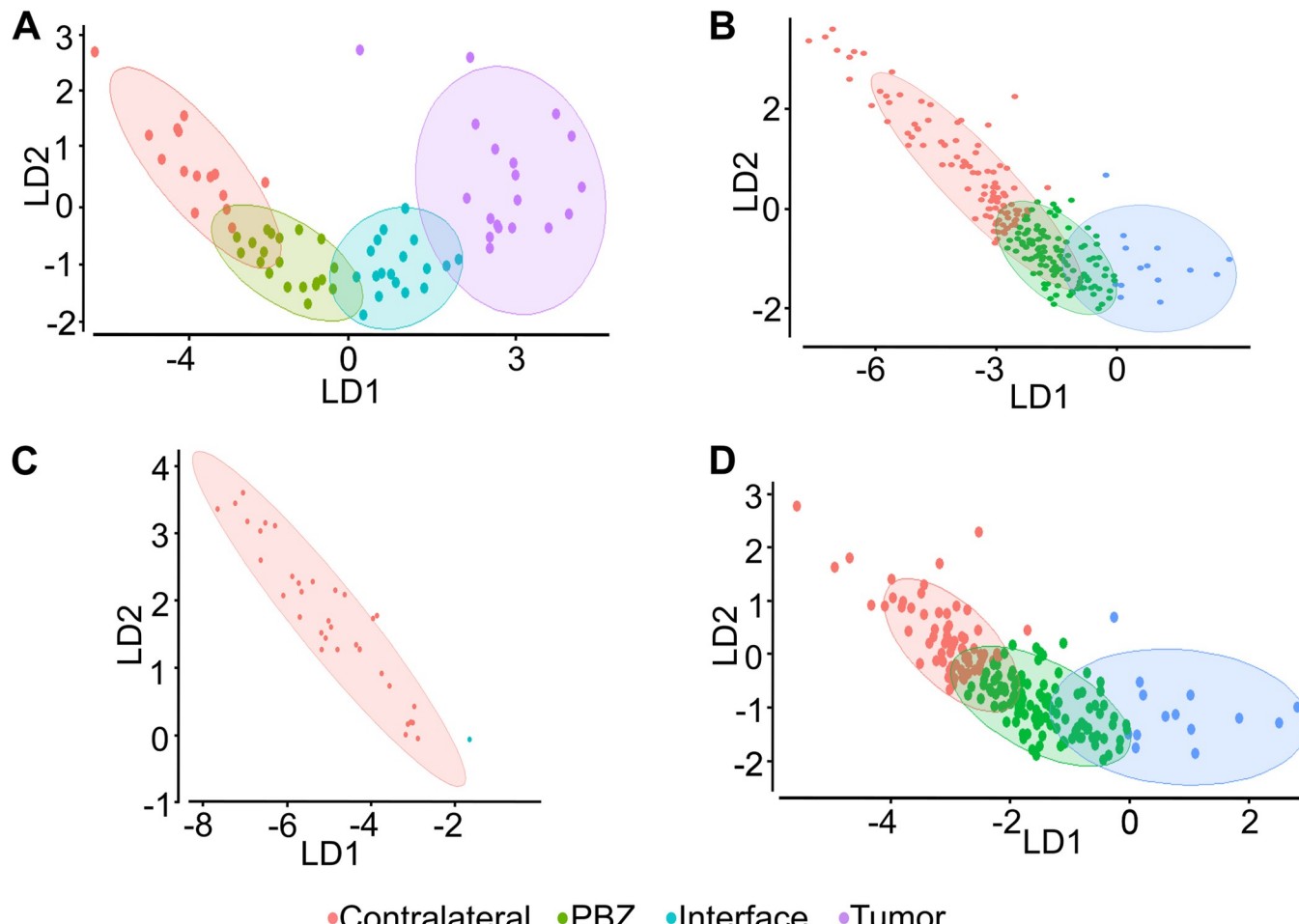

**Fig 5. Linear discriminant analysis classifies microglia according to their origin and reveals microglial heterogeneity in PBZ. (A)** Plot of the linear discriminants 1 and 2 (LD1 and LD2, respectively) obtained after training a linear discriminant function with the cell area, density and circularity of 171 cells from the four regions of interest (ROIs) and tested with 72 cells also from the four ROIs (accuracy: 94.4%). Ellipses represent the 95% confidence interval for each cluster. **(B)** Plot of LD1 and LD2 obtained with the same linear discriminant function trained in **(A)** and tested with 200 PBZ microglia and 36 control microglia (accuracy: 62.7%). **(C)** Plot of LD1 and LD2 obtained with the same linear discriminant function trained in **(A)** and tested with the 36 control microglia (accuracy: 97.2%). **(D)** Plot of LD1 and LD2 obtained with the same linear discriminant function trained in **(A)** and tested with the 200 PBZ microglia (accuracy: 56.5%). Note the low accuracy of this test and the fact that some PBZ cells were wrongly classified as control microglia and as those from the interface area (none was classified as tumoral microglia).

were correctly classified and only one was classified as PBZ microglia (Fig 5C). However, when testing the PBZ cells only 56.50% of them were correctly classified, while the rest of them were classified as belonging either to the CL (73 cells) or to the INT (14 cells) zones (Fig 5D; no cell was classified as tumoral). Considering the definite accuracy of LDA classifier in the CL set, these results indicate that PBZ cells are not homogeneous and could include different morphotypes (closer to CL and INT morphotypes), as will be evaluated later (Fig 7).

## Principal component analysis can differentiate microglia from the tumor, its surroundings, the PBZ and control tissue, using the parameters not used in LDA

The results with LDA confirmed that PBZ microglia can be differentiated from the rest of the tumor-related microglial populations, and even control microglia, using only three

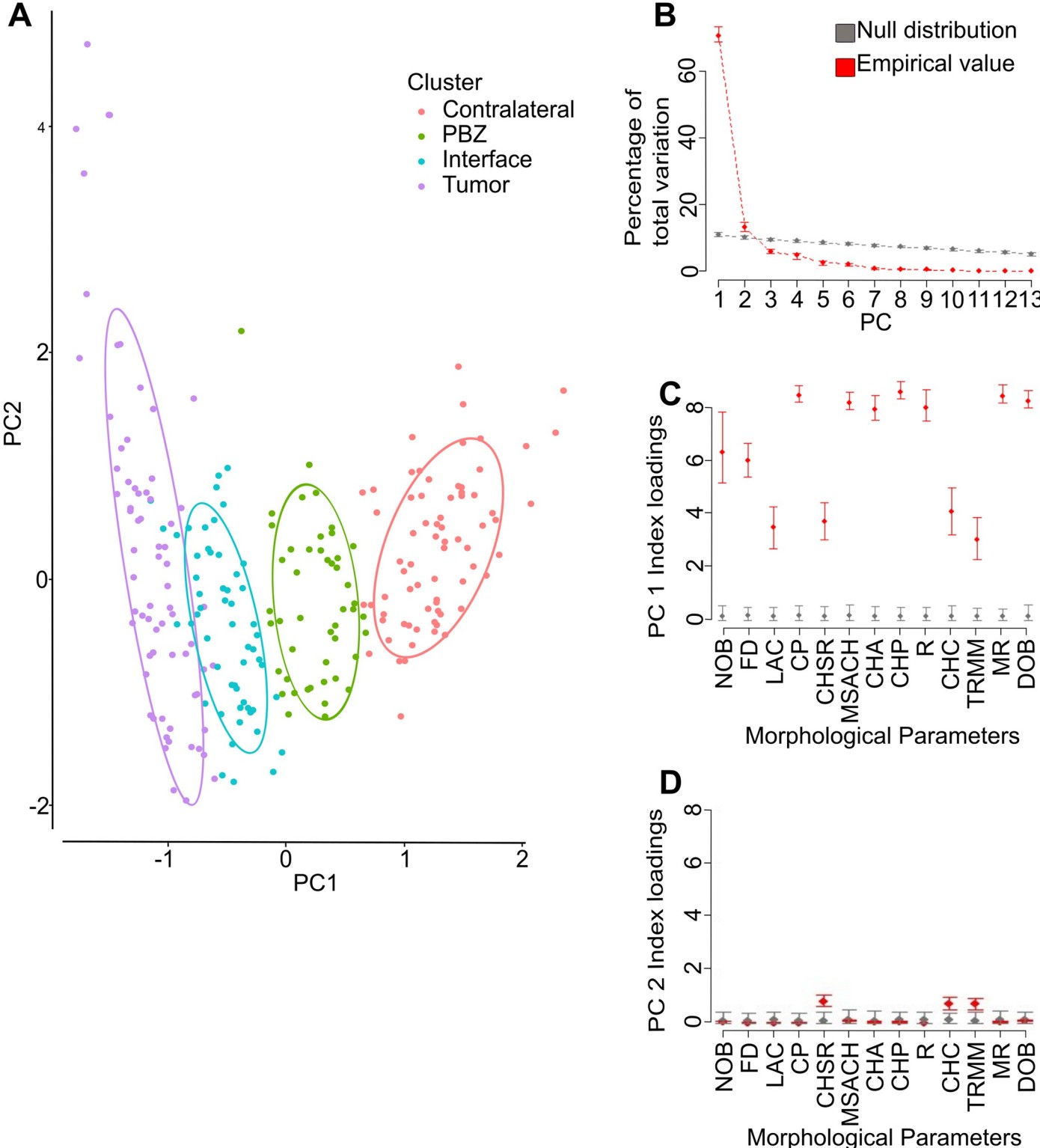

**Fig 6. Principal component analysis (PCA) classifies microglia according to their origin.** (A) Plot of the principal components 1 and 2 (PC1 and PC2, respectively), obtained after PCA using the 13 morphological parameters not used in LDA, of the 243 microglial cells from the four regions of interest. Ellipses represent the interval 95% confidence interval for each cluster. (B) Proportion of total variation contributed by the first PCs. Note that only PC1 and PC2 contributed significantly to variance beyond the null distribution. (C) Contribution of each morphological parameters to PC1 definition for the actual values and for the null distribution. (D) Contribution of each morphological parameters to PC2 definition for the actual values and the null distribution. Note that while all

parameters significantly contributed to PC1 definition, the convex hull span ratio (CHSR), Maximum/minimum Convex Hull radius ratio (TRMM) and the convex hull circularity (CHC) significantly contributed to PC2. Number of branches (NOB); Fractal dimension (FD); Lacunarity (LAC); Convex hull area (CHA); Cell perimeter (CP); Maximum span across the convex hull (MSACH); Convex hull perimeter (CHP); Roughness (R); Mean radius (MR); Diameter of the bounding circle (DOB).

morphological parameters (CA, DEN, and CC), two of them with high MMI. To further explore the robustness of this morphological clustering, we performed PCA with the remaining 13 parameters (Fig 6). With this analytical approach we found that PC1 represented 70.60% of total variation (95% CI:68.90 to 73.30), while PC2 represented 13.10% of total variation (95% CI:11.80 to 14.60; Fig 6B). When plotting the cells in the PCA1 and PCA2 dimensions, the four clusters previously identified, with a similar pattern as the one found in LDA, with TUM and CL cells located at opposite ends of the PC1-2 space, INT cells being closer to TUM cell while PBZ being closer to CL cells (Fig 6A). When assessing the contribution of each parameter to the definition of PC1 (Fig 6C) we found that all parameters significantly contributed to varying degrees, but CHP (Index load:8.55), CP (Index load:8.43), MR (Index load:8.41), DOB (Index load:8.22) and MSACH (Index load:8.16) exhibited the higher contributions to PC1. In contrast, when assessing the contribution of each parameter to the definition of PC2 (Fig 6D) we found that only CHSR (Index load:0.85), TRMM (Index load:0.76) and CHC (Index load:0.73) significantly contributed to its definition (Fig 6D).

## Morphological heterogeneity of PBZ microglia

As previously suggested by LDA, PBZ microglia could constitute a heterogeneous population (Fig 5). Thus, to test this possibility, we performed HCA for the independent sample of 200 PBZ cells (Fig 7A). The Silhouette method and the Calinski-Harabasz Index (Fig 7A insets) indicated that PBZ population could be segregated into two clusters (Fig 7A inset). Then, PCA

**Table 3. Mean and standard deviation of the 16 morphological parameters of the two clusters of peritumoral microglia along with their statistical comparison.**

| Parameter | Peritumoral Cluster 1 (n = 69) | Peritumoral Cluster 2 (n = 131) | Welch´s t-test (df) |
|---|---|---|---|
| NOB | 90.29 ± 38.17 | 151.85 ± 48.47 | t(168.84) = -9.85 p< 2.2e-16 |
| FD | 1.41± 0.07 | 1.44 ± 0.05 | t(107.47) = -3.13 p = 2.25e-3 |
| LAC | 1.08 ± 0.24 | 1.27 ± 0.30 | t(164.05) = -4.73 p = 4.76e-6 |
| CA(μm$^2$) | 90.29 ± 29.58 | 163.45 ± 44.74 | t(187.9) = -13.84 p< 2.2e-16 |
| DEN | 0.17 ± 0.06 | 0.13 ± 0.03 | t(86.55) = 5.85 p = 8.49e-8 |
| CP(μm) | 482.17 ± 156.26 | 945.39 ± 243.22 | t(190.24) = -16.32 p< 2.2e-16 |
| CHSR | 1.67 ± 0.44 | 1.41 ± 0.25 | t(92.47) = 4.43 p = 2.60e-5 |
| MSACH(μm) | 36.42 ± 6.98 | 52.49 ± 7.88 | t(153.67) = -14.79 p< 2.2e-16 |
| CHA (μm$^2$) | 572.45± 217.38 | 1336.24 ± 378.51 | t(196.49) = -18.11 p< 2.2e-16 |
| CHP(μm) | 93.63 ± 17.91 | 140.24 ± 19.09 | t(146.29) = -17.10 p< 2.2e-16 |
| R | 5.10 ± 1.31 | 6.70 ± 1.24 | t(131.69) = -8.37 p = 7.20e-14 |
| CC | 6.61e-3 ± 6.5e-3 | 2.54e-3 ± 1.1e-3 | t(69.95) = 5.09 p = 2.85e-6 |
| CHC | 0.79 ± 0.09 | 0.84 ± 0.05 | t(93.79) = -4.54 p = 1.67e-5 |
| TRMM | 1.79 ± 0.46 | 1.59 ± 0.24 | t(87.06) = 3.42 p = 9.64e-4 |
| MR(μm) | 15.85 ± 2.94 | 23.45 ± 3.34 | t(154.65) = -16.54 p< 2.2e-16 |
| DOB(μm) | 36.68± 7.05 | 52.99 ± 7.79 | t(150.75) = -14.16 p< 2.2e-16 |

Number of branches (NOB); Fractal dimension (FD); Lacunarity (LAC); Cell area (CA); Convex hull area (CHA); Density (DEN); Cell perimeter (CP); Convex hull span ratio (CHSR); Maximum span across the convex hull (MSACH); Convex hull perimeter (CHP); Roughness (R); Cell circularity (CC); Convex hull circularity (CHC); Maximum/minimum convex hull radius ratio (TRMM); Mean radius (MR); Diameter of the bounding circle (DOB).

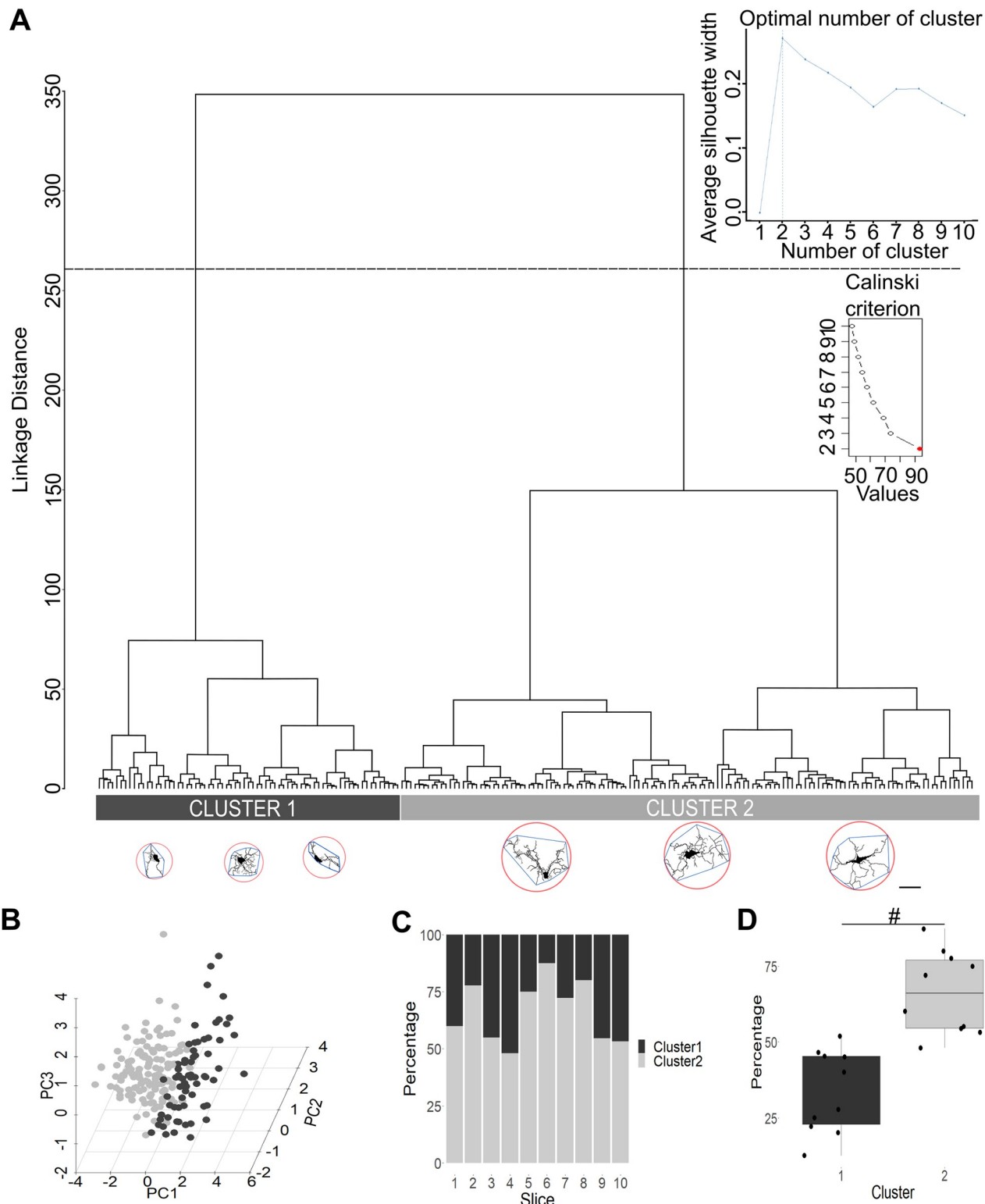

**Fig 7. Peritumoral brain zone (PBZ) microglia are heterogeneous. (A)** Dendrogram obtained with hierarchical clustering analysis, using the 16 morphological parameters of 200 PBZ microglial cells (represented in the X axis; three representative examples of each morphological cluster are presented; Scale bar = 10 μm; bounding circle is highlighted in red, and convex hull in blue) evaluating their Manhattan Distance (Y axis). The dashed line indicates the cut-off level to obtain two clusters, signaled as optimal trough the Silhouette method and the Calinski-Harabasz (Insets). **(B)** Plot of the principal components 1 and 2 (PC1 and PC2, respectively), obtained after PCA using the 16 morphological parameters of the 200 PBZ

microglia. Microglia belonging to cluster 1 and cluster 2 are represented in light gray and dark gray, respectively. **(C)** Proportional composition of PBZ microglia belonging to cluster 1 or 2 in all sampled slices. Note that both clusters are present in similar proportions in all sampled slices. **(D)** Statistical comparison between the percentage of cells belonging to cluster 1 and cluster 2 (n = 10 slices). #P<0.001.

was performed in this population with the 16 morphological parameters (Fig 7B). Three PCs represented the most significant proportion of data variability: PC1 represented 52.70% of total variation (95% CI:49.7 to 57.1; S4A Fig); PC2 represented 16.20% (95% CI:13.7 to 19.1; S4A Fig) of total variation and PC3 represented 12.90% (95% CI:10.7 to 14.2; S4A Fig) of the total variation. All 16 parameters significantly contributed to PC1, but CHP (Index loading: 7.59; S4A Fig), CP (Index loading: 7.49; S4B Fig), CHA (Index loading: 7.46; S4B Fig) and MR (Index loading: 7.12; S4B Fig) contributed the most. Seven parameters contributed to PC2 but CHC (Index loading:1.32; S4C Fig), TRMM (Index loading:1.14; S4C Fig) and CHSR (Index loading:1.08; S4C Fig) contributed the most. Finally, six parameters contributed to PC3 but FD (Index loading:0.92; S4D Fig) and LAC (Index loading:0.63; S4D Fig) contributed the most. PBZ microglial cells were separated in two major groups (Fig 7B and 7C) and the same cell subpopulation dominated (around two thirds) in each of them (Fig 7D): cluster one represented 33.65 ± 13.67% of PBZ cells while cluster two represented 66.35 ± 13.67% of them (t (18) = -5.35, P = 0.44 x $10^{-4}$). The values of all morphological parameters exhibited by these PBZ morphological clusters (Table 3) clearly indicate that one cluster exhibit values closer to the ones shown by control microglia (Table 1) while the other cluster presented values closer to the ones shown be INT microglia (Table 1).

## Discussion

Histological analysis of the GBM has revealed the presence of highly anaplastic and mitotic tumor cells immerse in a TMI mainly composed of infiltrating monocytes/macrophages (up to 50% of cell population), astrocytes, neural stem/progenitor cells, other immune infiltrates and, remarkably, brain-resident microglia [81, 82]. GBM-associated microglia facilitate glioma cell expansion [21–23], while their depletion induces glioma shrinkage [24]. Beyond the tumor, PBZ microglia contribute to GBM-induced seizure activity [83–87] and participate in creating an area prone for tumor recurrence [20, 34, 35]. Thus, it has become relevant to characterize the morphological features of PBZ microglia and compare them with microglia from the tumor and its immediate surroundings (as well as with control microglia), which we did in this study. A study by Milior et al., (2020) indicated that microglia localized in the peritumoral cortex are more ramified than those close to the tumor. Recently, Noorani et al., (2023) have shown that microglial motility and activation markers (Iba1 and CD68) are reduced in the invasive margins compared with the GBM core, while those of homeostatic or even anti-inflammatory microglia (i.e. P2Y12) are increased. However, a systematic morphological analysis and classification was required.

The C6 cell line has been considered "the gold standard in glioma research" [57], because it reproduces GMB high growth rate and vascularization [57]. C6 cells also express most markers found in human glioblastomas [57]. Certainly, GBM and their cell lines are heterogeneous [14, 39, 57, 88] and different GBM molecular identities differentially regulate microglial densities and phenotypes [88]. For instance, GMB overexpressing the epidermal growth factor receptor variant III (EGFRvIII) favors microglial infiltration [88]. This should be also the case for C6 cells, since they also express EGFRvIII [14], along with other genes such as PDGFβ, (IGF)-1, wild type EGFR, and Erb3/Her3 precursor proteins (for reviews see [57, 89]). C6 also have a reduced expression of IGF-2, FGF-9, and FGF-10 while there is no change in the expression of

MMP-7 gene (for reviews see [57, 89]). A genomic comparison indicates that C6 cells exhibit high similarity with mesenchymal GBMs, employing similar immune evasion strategies [39].

Based on previous studies in humans, we used gadolinium-based MRI to identify the tumor, its surroundings, the PBZ [42, 87, 90, 91], and the ROIs, and to thoroughly characterize the microglia morphology in these areas and the contralateral region, as a control. This regional differentiation was facilitated by the fact that the tumors in our study were discrete and well-demarcated, as those previously described using the same tumoral cells [42, 54–57]. However, advance human GBMs are more infiltrative [57] and, thus, our results could only be extrapolated to early human GBMs [54–57]. We mainly used clustering and discriminative approaches that have been previously reported and validated [9, 15–17, 68, 92, 93]. Interestingly, we found that most objective studies often identified four general microglia morphotypes [9, 11, 15, 17, 93], which could be related to the classical morphotypes of ramified microglia, hyper-ramified reactive phenotype, the classical 'reactive' morphotype and the 'phagocytic' morphology [5–7].

Here, we used a combination of morphological parameters of microglia that have been previously validated for discrimination of different microglial subpopulations [9, 11, 15, 17, 68, 93] and found that some of these parameters are not only particularly useful to classify PBZ microglia and separate them from microglia in the tumor, its surroundings and control tissue, but also to reveal previously described microglial morphotypes [9, 72]. For instance, the high FD values in the contralateral microglia (i.e. control microglia), which are shared by interface and PBZ microglia, are similar to those already reported in control tissue and correspond to ramified surveillant microglia [9, 72]. In contrast, low FD exhibited by microglia in the tumor resemble the low FD found in lesioned tissue and correspond to unramified/ameboid microglia [9, 94]. Similarly, tumor microglia also exhibited low LAC values revealing a more compact cell type [9, 70, 72], in contrast to higher LAC values found in control and, remarkably, in PBZ microglia, which are related to more extended and ramified microglia [9, 70, 72]. Not only were the FD and LAC values exhibited by PBZ microglia similar to those found in control microglia (Fig 3; S1 Fig), but both morphotypes shared TRMM and DEN values (Fig 3; S1 Fig; Table 1). However, PBZ microglia exhibited differences in a variety of parameters (i.e., CHA, CP, NOB, MSACH) compared to control microglia (Fig 3; S1 Fig; Table 1), clearly indicating that PBZ microglia can still be differentiated from control tissue microglia, and one PBZ subpopulation exhibited morphological characteristics closer to the ones exhibited by INT microglia (Table 3).

Most morphological parameters, excepting LAC, CA, DEN, TMRR, CHC and CHSR, exhibit gradual changes when microglia are characterized from tumoral areas towards control tissue or vice versa (Fig 3; S1 Fig; Table 1). Interestingly, CP shows a gradual decrease from the maximal values found in control tissue to minimal values found within the tumor, and with intermediate values in the PBZ (Fig 3; S1 Fig; Table 1), indicating that, as shown by others [65, 66, 95], proinflammatory conditions lead to a decrease in CP and might persist in the PBZ. Other morphological parameters, including NOB and CHA (Fig 3; S1 Fig; Table 1) show a gradual change as the sampled tissue moved from control tissue towards the tumoral zone. These gradual morphological transitions in specific morphological parameters preclude the clear definition of any microglial morphotype or to the differentiation of PBZ microglia from microglia of other regions. Thus, multivariate analyses seem to be necessary to achieve such categorizations [9, 11, 15, 17, 93].

HCA has proven to be useful, especially in our experimental conditions, to classify microglial subpopulations based on different morphometric parameters ([9, 15, 17, 68, 92]; Fig 4). Furthermore, the robustness of this classification was confirmed, preceded by MMI assessment [9, 15–17, 92], with LDA using only three parameters (CA, DEN, and CC; Fig 5). Interestingly,

the pattern of cell distribution in the graphical representation of the LDA shows a "trajectory" of control cells on one end of the distribution, and in close proximity to the PBZ transitioning into the INT cell type and reaching the tumoral microglial morphotype at the opposite end of the distribution, which is a pattern also observed with PCA (Figs 5 and 6). This type of morphological trajectory, of gradual and continuous morphological changes, has also been described in other pathological conditions [9, 15, 17]. Of notice, the accuracy of LDA was extremely high (94.4%) and the two main lineal discriminants explained most of the total variance. DEN was the strongest predictor of the LD1 function, and CA was the strongest predictor for LD2. As previously shown, the DEN values found in tumor microglia resemble those in microglia under proinflammatory conditions [9, 65] or within damaged tissue [64, 66–68]. In contrast, the low CA values found in the tumor (compared with higher value found in control tissue and the PBZ) are comparable to those reported under proinflammatory conditions [9, 15] or damaged tissue [67, 68], as well. Interestingly, CA values cannot be differentiated between tumor microglia and those from its surroundings, but they are significantly different in PBZ microglia compared to all evaluated areas. In contrast, although DEN values cannot be differentiated between control and PBZ microglia, these DEN values are significantly different from those found in the tumor and its surroundings, making the combination of both values a strong tool to differentiate microglia morphotypes [9, 68], perhaps because CA and DEN parameters are poorly correlated (Fig 4; [9, 68]).

As others have done [9, 15], to further corroborate the robustness of microglial classification found with HCA and LDA, PCA can also be used to differentiate microglial morphotypes using the remaining morphological parameters not included in the LDA [9]. We found that this is the case, indicating that the microglial classification revealed in our study is robust and that different morphological parameters fed to different analytical approaches render almost identical microglial classifications and a very similar graphical pattern (Figs 5 and 6). In the PCA analysis we identified that CP, CHP and CHA contributed the most to PC1 definition, while PC2 was most influenced by CHSR, CHC and TRMM. Interestingly, all the morphological parameters that mostly define PC1 (CP, CHP, CHA) exhibit a graded change from high values in control microglia to reduced values in the tumor, and with PBZ exhibiting intermediate values closer (but significantly different) to those of control microglia (Fig 3; S1 Fig; Table 1). The reduction in these parameters has already been observed in proinflammatory conditions [9, 70] and under cell damage [68, 69], compared with control tissue. In contrast, while two of the parameters that contributed to PC2 definition (CHSR and TRMM), exhibited a graded change from control microglia towards the tumor, they also showed an opposite change to the parameters that defined PC1, with low values in control tissue, high values in the tumor and the PBZ exhibiting intermediate values closer to those of control microglia (Fig 3; S1 Fig; Table 1). The increase in CHSR and TRMM has also been observed in proinflammatory conditions [9] although both parameters do not change under cell damage conditions [69]. Altogether, the results of this study support that PBZ microglia exhibit a morphological profile clearly differentiable to that found in CL, INT and TUM zones. However, taking the analytical tools applied in this study to the extreme, we also found that PBZ is not a homogeneous population, but that can be segregated into two subsets of cells, using HCA and PCA, with one being more abundant (about two thirds) than the other (one third). Several morphological parameters already discussed contributed to the definition of the principal components that allowed the identification of these PBZ subpopulations, such that PC1 was mostly defined by CHP, CP, CHA and MR. PC2 was mostly defined by CHC, TRMM and CHSR, whereas PC3 was mostly defined by FD and LAC. Future evaluations will provide the specific morphological features that define these PBZ subpopulations. However, morphological parameters exhibited by one morphological subpopulation of the PBZ (Table 3) are closer to the ones shown in

control microglia (Table 1), while the other subpopulation exhibit values closer to the ones shown by INT microglia (Table 1), which indicate that this PBZ microglia segregation would be part of the morphological transition (continuum) from control to tumoral morphotypes.

Aside from the objective microglial classification and differentiation of PBZ microglia, our study corroborated that one microglial morphotype identified in the contralateral tissue (control microglia) is characterized by a high FD, LAC, CA and CP along with low DEM [9, 64–70]. As already mentioned, these control microglia are not different from the one observed in the absence of tumor (S3 Fig). In contrast, the other microglial morphotype described in this study (tumor microglia) exhibited low FD, LAC, CA and CP along with high DEN, which resemble the ameboid/activated microglia [9, 64–70]. These results are in line with several previous reports describing tumoral microglial morphology as amoeboid [19, 96–100]; however, transcriptional analysis showed that microglia exhibit both M1 and M2 phenotypes within murine brain tumors. Thus, these cells present a more complex biology [101, 102]. Of relevance, PBZ microglia show several characteristics similar to those of control cells, such as high FD and LAC along with low DEN [9, 64–70]. However, our characterization also showed that PBZ microglia also exhibit intermediate morphological values (i.e. CA, CP, DEN, MASCH, MR, DOB, CHA and NOB) between control and tumor cells, which would reflect a transition morphotype that could support tumor recurrence [20, 34, 35]. Thus, it is likely that PBZ microglia release factors that modify the extracellular matrix degradation which would favor glioma cell reemergence [20, 34–36] and also contribute to the generation of excitability conditions that worsen the symptomatology induced by glioma recurrence [83–87].

## Supporting information

**S1 Fig. Comparison of the 16 morphological parameters of microglia sampled in the four regions of interest (T = tumor, I = interface, PBZ = peritumoral, CL = contralateral hemisphere).** Number of Branches (NOB); Fractal dimension (FD); Lacunarity (LAC); Cell Area (CA); Convex Hull Area (CHA); Density (DEN); Cell perimeter (CP); Convex Hull Span Ratio (CHSR); Maximum span across the Convex Hull (MSACH); Convex Hull Perimeter (CHP); Roughness (R); Cell circularity (CC); Convex Hull Circularity (CHC); The ratio maximum/minimum Convex Hull radii (TRMM); Mean radius (MR); Diameter of the Bounding Circle (DOB). *P<0.05, **P<0.01, #P<0.001 (n = 80 cells/group).
(DOCX)

**S2 Fig. Multivariate comparisons among the sixteen morphological parameters measured in microglia sampled from the four ROIs.** Significance matrix was built for the following parameters: Number of branches (NOB); Fractal dimension (FD); Lacunarity (LAC); Cell area (CA); Convex hull area (CHA); Density (DEN); Cell perimeter (CP); Convex hull span ratio (CHSR); Maximum span across the convex hull (MSACH); Convex hull Perimeter (CHP); Roughness (R); Cell circularity (CC); Convex hull circularity (CHC); Maximum/minimum convex hull radius ratio (TRMM); Mean radius (MR); Diameter of the bounding circle (DOB) comparing the four regions of interest (T = tumor, I = interface, PBZ, CL = contralateral hemisphere). P-values are provided and represented in a blue scale (n = 80 cells/group).
(DOCX)

**S3 Fig. Comparison of the 16 morphological parameters of microglia sampled in two regions (CL = contralateral hemisphere, WOTUMOR = without tumor, SH = sham).** Number of Branches (NOB); Fractal dimension (FD); Lacunarity (LAC); Cell Area (CA); Convex Hull Area (CHA); Density (DEN); Cell perimeter (CP); Convex Hull Span Ratio (CHSR); Maximum span across the Convex Hull (MSACH); Convex Hull Perimeter (CHP); Roughness

(R); Cell circularity (CC); Convex Hull Circularity (CHC); The ratio maximum/minimum Convex Hull radii (TRMM); Mean radius (MR); Diameter of the Bounding Circle (DOB). *$P<0.05$, #$P<0.001$.
(DOCX)

**S4 Fig. Parameters defining principal component analysis (PCA) of PBZ microglia. (A)** Proportion of total variation contributed by the first principal components (PC). Note that only PC1, PC2 and PC3 contributed significantly to variance beyond the null distribution. **(B)** Contribution of each morphological parameters to PC1 definition for the actual values and for the null distribution. **(C)** Contribution of each morphological parameters to PC2 definition for the actual values and for the null distribution. **(D)** Contribution of each morphological parameters to PC3 definition for the actual values and for the null distribution. Number of Branches (NOB); Fractal dimension (FD); Lacunarity (LAC); Cell Area (CA); Convex Hull Area (CHA); Density (DEN); Cell perimeter (CP); Convex Hull Span Ratio (CHSR); Maximum span across the Convex Hull (MSACH); Convex Hull Perimeter (CHP); Roughness (R); Cell circularity (CC); Convex Hull Circularity (CHC); The ratio maximum/minimum Convex Hull radii (TRMM); Mean radius (MR); Diameter of the Bounding Circle (DOB).
(DOCX)

## Acknowledgments

The authors would like to thank Dr. Luis Concha, Dr. Juan José Ortíz Retana (Lanirem-UNAM), Dr. Pavel Rueda, Dr. Aleph Prieto, Citlali Suárez Rangel, M. Sc. Azucena Ruth Aguilar Vázquez and Dr. Benito Ordaz for technical support; Dr. Sofía Díaz-Cintra for her support in image acquisition; Anaid Antaramian and and Jessica González for editing the manuscript.

## Author Contributions

**Conceptualization:** G. Anahí Salas-Gallardo, Fernando Peña-Ortega.

**Data curation:** G. Anahí Salas-Gallardo.

**Formal analysis:** G. Anahí Salas-Gallardo, Jonathan-Julio Lorea-Hernández, Ángel Abdiel Robles-Gómez.

**Investigation:** G. Anahí Salas-Gallardo.

**Methodology:** Jonathan-Julio Lorea-Hernández, Ángel Abdiel Robles-Gómez, Fernando Peña-Ortega.

**Project administration:** Fernando Peña-Ortega.

**Resources:** Claudia Castillo-Martin Del Campo, Fernando Peña-Ortega.

**Supervision:** Claudia Castillo-Martin Del Campo, Fernando Peña-Ortega.

**Writing – original draft:** G. Anahí Salas-Gallardo.

**Writing – review & editing:** G. Anahí Salas-Gallardo, Claudia Castillo-Martin Del Campo, Fernando Peña-Ortega.

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
