## [Decision Letter · Decision Letter 0]

6 Sep 2023

PONE-D-23-16668Morphological differentiation of peritumoral brain zone microglia.PLOS ONE

Dear Dr. Peña,

Thank you for submitting your manuscript to PLOS ONE. After careful consideration, we feel that it has merit but does not fully meet PLOS ONE’s publication criteria as it currently stands. Therefore, we invite you to submit a revised version of the manuscript that addresses the points raised during the review process. Importantly, a control sham group is required to validate your findings.

We look forward to receiving your revised manuscript.

Kind regards,

Giuseppe Biagini, MD

Academic Editor

PLOS ONE

“G.A.S.G. received a graduate fellowship from CONACyT (I.D. 770620). This work was supported by Dirección General de Asuntos del Personal Académico UNAM (Grant IG200521) and CONACyT (A1-S-7540).”

“The authors would like to thank Dr. Luis Concha, Dr. Juan José Ortíz Retana (Lanirem-UNAM), M. Sc. Azucena Ruth Aguilar Vázquez and Dr. Benito Ordaz for technical support; Dr. Sofía Díaz-Cintra for her support in image acquisition and Jessica González for editing the manuscript. G. Anahí Salas-Gallardo received a graduate fellowship from CONACyT (I.D. 770620). This work was supported by Dirección General de Asuntos del Personal Académico UNAM (Grant IG200521) and CONACyT (A1-S-7540).”

“G.A.S.G. received a graduate fellowship from CONACyT (I.D. 770620). This work was supported by Dirección General de Asuntos del Personal Académico UNAM (Grant IG200521) and CONACyT (A1-S-7540).”

7. Please remove your figures from within your manuscript file, leaving only the individual TIFF/EPS image files, uploaded separately. These will be automatically included in the reviewers’ PDF.

8. We note that Figures 1 and 2 in your submission contain copyrighted images. All PLOS content is published under the Creative Commons Attribution License (CC BY 4.0), which means that the manuscript, images, and Supporting Information files will be freely available online, and any third party is permitted to access, download, copy, distribute, and use these materials in any way, even commercially, with proper attribution. For more information, see our copyright guidelines: http://journals.plos.org/plosone/s/licenses-and-copyright.

1. You may seek permission from the original copyright holder of Figures 1 and 2 to publish the content specifically under the CC BY 4.0 license.

9. We notice that your supplementary figures are included in the manuscript file. Please remove them and upload them with the file type 'Supporting Information'. Please ensure that each Supporting Information file has a legend listed in the manuscript after the references list.

Reviewers' comments:

Reviewer's Responses to Questions

**Comments to the Author**

1. Is the manuscript technically sound, and do the data support the conclusions?

Reviewer #1: Yes

Reviewer #2: Partly

2. Has the statistical analysis been performed appropriately and rigorously? 

Reviewer #1: Yes

Reviewer #2: No

3. Have the authors made all data underlying the findings in their manuscript fully available?

Reviewer #1: Yes

Reviewer #2: No

4. Is the manuscript presented in an intelligible fashion and written in standard English?

Reviewer #1: Yes

Reviewer #2: Yes

5. Review Comments to the Author

Reviewer #1: This manuscript by Pena et al looks at the peritumoural brain zone to determine whether microglial morphology can be reliably distinguished from other tumour areas. They employed the rat C6 transplant model and assessed microglial morphology from these rat tumours, showing with hierarchical clustering analysis, LDA and PCA that there was a separate cluster of PBZ microglia compared with other tumour areas. Although there was some overlap seen in the clusters as would be expected given gliomas have heterogeneous cell populations, the PBZ microglia do seem to form a separate morphological cluster. This is a useful study that contributes to a growing literature on the importance of PBZ microglia for glioblastoma biology. However, these concerns need to be addressed:

1. In Figure 1, we see that the rat gliomas are very discrete and not as infiltrative as typical human GBMs. This must be highlighted and discussed. Moreover, how these results may translate to human GBMs given this caveat needs further discussion.

2. Further discussion of relevant literature is needed eg recent studies have corroborated reduced PBZ microglia compared to the GBM core, with differences in microglial markers between these regions including in pro-inflammatory and anti-inflammatory markers - these works need to be discussed in context of the findings presented here - PMID 37324244

3. Fig 1c is very low resolution - needs improved version.

4. Fig 3 - the lettering / numbers are difficult to read - needs clarification. This is also true for Fig 4B, and supplementary figures 1 and 2.

5. It is unclear what subtype of GBM the authors are trying to model - it is known that different GBM molecular subtypes may contain different microglial patterns eg EGFR amplified vs non-amplified eg PMID: 30401716. Some discussion of this is warranted.

Reviewer #2: This study describes some morphological differences between microglial cells located within the glioblastoma, at interface, and within the adjacent area. Potentially, the results are of significant clinical importance. There are several technical concerns regarding the study.

The number of rats used for each specific purpose, mortality rates, etc should be given in methods. Why authors did not create a sham surgery group in which nontumoral cell could be injected using identical procedures. It is important, since authors then could compare the effects of tumor with the effects of trauma induced by the needle. In fact, the main goal of authors is to describe microglia in the peritumoral zone, but with this design the changes reported can be caused by this mild trauma.

“Then, it was transferred to a 30% sucrose solution until saturation.”

How authors define the saturation?

Non-specific binding is blocked by normal (bovine) serum, while tween and Triton X-100 are optional components of a solvent. Please, rephrase.

Please, specify manufacturer and reference for the secondary antibody.

Please, describe the Nissl technique used.

“Lemée et al. (2015) defined the PBZ as a radiologically normal peritumoral area, located within a 2 cm distance from the brain/tumor interface….”

“Then, MRI image was binarized and 2 mm from the tumor border were measured…”

Please, explain this transition from 2 cm to 2 mm.

“To quantify the number of microglia in a 40x photomicrographs (25.96 x 103 μm2), seven slices (from four animals) were sampled to…”

This looks more like a density measurement. To quantify the number… one would need to use stereological counting and, in any case, 7 slices from 4 animals would not be sufficient.

“Eight cells per area (tumoral, inteface, peritumoral and contralateral), imaged at the highest magnification, were selected per slice.”

Please, give the criteria of the selection. These criteria should guarantee true randomness.

In statistics, why different tests (including normality tests) were used in each specific dataset should be explained. Many, if not all, morphological indices look intercorrelated. Therefore, MANOVA (or an equivalent test for nonparametric data) should be used to correct for multiple comparisons. For the very same reasons, the use of the regression analysis may be questionable. T-test may be too weak for post-hoc comparisons.

6. PLOS authors have the option to publish the peer review history of their article (what does this mean?). If published, this will include your full peer review and any attached files.

Reviewer #1: No

Reviewer #2: No

---

## [Author Response · Author response to Decision Letter 0]

21 Dec 2023

Responses

Editor:

Importantly, a control sham group is required to validate your findings.

Answer (A): We have incorporated the requested group and found that sham microglia are mostly undistinguishable from contralateral microglia and/or from microglia of tissue in which tumor was not developed (despite C6 cells were injected).

We note that you have included the phrase “data not shown” in your manuscript. Unfortunately, this does not meet our data sharing requirements. 

A: We have deleted the expression “data not shown” and included the required information.

We note that Figures 1 and 2 in your submission contain copyrighted images.

A: Figures 1 and 2 DID NOT AND DO NOT contain copyrighted images.

Reviewer #1: 

1. In Figure 1, we see that the rat gliomas are very discrete and not as infiltrative as typical human GBMs. This must be highlighted and discussed. Moreover, how these results may translate to human GBMs given this caveat needs further discussion.

A: Yes, the tumors were discrete and well-demarcated, likely due to the reduced number of injected cells and the host strain used (San-Galli et al., 1989), as has been found by others (Inoue et al. 1987, Farrell et al., 1987; San-Galli et al., 1989; Doblas et al., 2010; Giakoumettis et al., 2018). Indeed, magnetic resonance imaging show that C6 gliomas have a very clear sharpness in the transition from tumor to surrounding zones (Doblas et. al., 2010). While this actually was an advantage for our study, allowing us to consistently define the different zones for microglial characterization, we have acknowledged that the tumors obtained in this study do not exactly resemble advanced human GBMs (Inoue et al. 1987, Farrell et al., 1987; San-Galli et al., 1989; Giakoumettis et al., 2018). Perhaps our tumors only resemble early human GBMs (Inoue et al. 1987, Farrell et al., 1987; San-Galli et al., 1989; Giakoumettis et al., 2018). This information has been included in the Materials and Methods and the Discussion.

2. Further discussion of relevant literature is needed eg recent studies have corroborated reduced PBZ microglia compared to the GBM core, with differences in microglial markers between these regions including in pro-inflammatory and anti-inflammatory markers - these works need to be discussed in context of the findings presented here - PMID 37324244.

A: Relevant new literature has been incorporated. For instance: Pyonteck et al., 2013; Gieryng et al., 2017; Landry et al. 2020; Akkari et al., 2020; Noorani et al., 2023; Petterson et al., 2023 (look at the reference list ate the end of the answers).

3. Fig 1c is very low resolution - needs improved version.

A: We improved the resolution of the panel.

4. Fig 3 - the lettering / numbers are difficult to read - needs clarification. This is also true for Fig 4B, and supplementary figures 1 and 2.

A: We improved the lettering / numbers in all Figures mentioned.

5. It is unclear what subtype of GBM the authors are trying to model - it is known that different GBM molecular subtypes may contain different microglial patterns eg EGFR amplified vs non-amplified eg PMID: 30401716. Some discussion of this is warranted.

A: In this study we used the C6 cell line, which has been considered “the gold standard in glioma research” (Giakoumettis et al., 2018), because it reproduces GMB high growth rate and vascularization (Giakoumettis et al., 2018). C6 cells also express most markers found in human glioblastomas (Giakoumettis et al., 2018). Certainly, GBM and their cell lines are heterogeneous (Sibenaller et al., 2015; Gieryng et al., 2017; An et al., 2018; Giakoumettis et al., 2018; Mingzhi Zhang et al., 2019) and different GBM molecular identities differentially regulate microglial densities and phenotypes (An et al., 2018). For instance, GMB overexpressing the epidermal growth factor receptor variant III (EGFRvIII) favors microglial infiltration (An et al., 2018). This should be also the case for C6 cells since they also express EGFRvIII (Zhang et al., 2019), along with other genes such as PDGFβ, (IGF)-1, wild type EGFR, and Erb3/Her3 precursor proteins (for reviews see Barth and Kaur 2009; Giakoumettis et al., 2018). C6 also have a reduced expression of IGF-2, FGF-9, and FGF-10 while there is no change in the expression of MMP-7 gene (for reviews see Barth and Kaur 2009; Giakoumettis et al., 2018). A genomic comparison indicates that C6 cells exhibit high similarity with mesenchymal GBMs employing similar immune evasion strategies (Gieryng et al., 2017). This information has been included in the Discussion. 

Reviewer #2: 

-The number of rats used for each specific purpose, mortality rates, etc should be given in methods.

A: We have provided the requested information in the materials and methods.

Why authors did not create a sham surgery group in which nontumoral cell could be injected using identical procedures. It is important, since authors then could compare the effects of tumor with the effects of trauma induced by the needle. In fact, the main goal of authors is to describe microglia in the peritumoral zone, but with this design the changes reported can be caused by this mild trauma.

A: In our previous submission we included two controls: 1) the contralateral tissue as well as 2) animals without a tumor which directly deal with the issues mentioned by the reviewer. However, based on the reviewer’s suggestion, we now we have included sham tissue, as requested (now in Supplementary Figure 3).

“Then, it was transferred to a 30% sucrose solution until saturation.”

How authors define the saturation?

A: We defined saturation when the brain sank to the bottom of the 30% sucrose solution (Peña and Tapia, 1999; 2000; Tashiro et al., 2011). This information was included in the Materials and Methods section. 

Non-specific binding is blocked by normal (bovine) serum, while tween and Triton X-100 are optional components of a solvent. Please, rephrase.

A: The sentence has been rephrased.

Please, specify manufacturer and reference for the secondary antibody.

A: The information has been included in Materials and Methods.

Please, describe the Nissl technique used. 

A: We used cresyl violet staining (Peña and Tapia, 1999; 2000; Lorea-Hernández et al., 2016; Mendez-Salcido et al., 2022) as now indicated in Materials and Methods.

“Lemée et al. (2015) defined the PBZ as a radiologically normal peritumoral area, located within a 2 cm distance from the brain/tumor interface….”

“Then, MRI image was binarized and 2 mm from the tumor border were measured…”

Please, explain this transition from 2 cm to 2 mm.

A: We based this transition on the proportion between human brain mass (1508.91 ± 299.14; Azevedo et al., 2009) and rat brain mass (1.802 ± 0.313; Herculano-Houzel et al., 2006), which is 0.00119 and very close to the proportion used by us. This information has been incorporated in the Material and Methods section.

“To quantify the number of microglia in a 40x photomicrographs (25.96 x 103 μm2), seven slices (from four animals) were sampled to…”

This looks more like a density measurement. To quantify the number… one would need to use stereological counting and, in any case, 7 slices from 4 animals would not be sufficient.

A: Agree and have indicated that what we actually measured was microglial density (as already stated in the Y-axis of Fig. 1E).

“Eight cells per area (tumoral, inteface, peritumoral and contralateral), imaged at the highest magnification, were selected per slice.”

Please, give the criteria of the selection. These criteria should guarantee true randomness.

A: For this purpose, the 40x objective was randomly directed to each of the zones. All entire and nonoverlapping cells within the randomly selected area were selected, imaged at 100x and analyzed. When the area imaged with the 40x objective included more than the required eight cells, a random number generator was used to select the ones to be incorporated in the sample. This information has been incorporated in the Material and Methods section.

In statistics, why different tests (including normality tests) were used in each specific dataset should be explained. Many, if not all, morphological indices look intercorrelated. Therefore, MANOVA (or an equivalent test for nonparametric data) should be used to correct for multiple comparisons. For the very same reasons, the use of the regression analysis may be questionable. T-test may be too weak for post-hoc comparisons.

A: We aimed to apply the more adequate statistical tests based on the specific characteristics of any data set. However, we agree with the reviewer and performed the suggested analysis. We obtained identical results by performing a Non-Parametric MANOVA (NPMANOVA) using the adonis function from vegan package in RStudio Software (Noorani et al., 2023). All 16 measured parameters were evaluated for the 4 ROIs and significative differences were found (F(3)=262.99, p=0.001). A wrapper function for multilevel pairwise comparison using the pairwise.adonis function with Bonferroni correction was used as post-hoc test. The following results were obtained: CL vs PBZ (F(1)= 97.339504, p=0.01), CL vs INT (F(1)= 470.320998, p=0.01), CL vs TUM (F(1)= 723.741213, p=0.01), PBZ vs INT (F(1)= 194.567536, p=0.01), PBZ vs TUM (F(1)= 426.054392, p=0.01), INT vs TUM (F(1)= 81.390667, p=0.01). Regarding T-test, we only use it to compare the two types of microglia described in Fig, 7, which we think is appropriate in this case. The MPMANOVA comparisons were included in the results and as new Supplementary figure 2.

References

Akkari L, Bowman RL, Tessier J, Klemm F, Handgraaf SM, de Groot M, Quail DF, Tillard L, Gadiot J, Huse JT, Brandsma D, Westerga J, Watts C, Joyce JA. Dynamic changes in glioma macrophage populations after radiotherapy reveal CSF-1R inhibition as a strategy to overcome resistance. Sci Transl Med. 2020;12(552):eaaw7843.

An Z, Knobbe-Thomsen CB, Wan X, Fan QW, Reifenberger G, Weiss WA. EGFR Cooperates with EGFRvIII to Recruit Macrophages in Glioblastoma. Cancer Res. 2018 Dec 15;78(24):6785-6794.

Azevedo FA, Carvalho LR, Grinberg LT, Farfel JM, Ferretti RE, Leite RE, Jacob Filho W, Lent R, Herculano-Houzel S. Equal numbers of neuronal and nonneuronal cells make the human brain an isometrically scaled-up primate brain. J Comp Neurol. 2009 Apr 10;513(5):532-41.

Barth RF, Kaur B. Rat brain tumor models in experimental neuro-oncology: the C6, 9L, T9, RG2, F98, BT4C, RT-2 and CNS-1 gliomas. J Neurooncol. 2009 Sep;94(3):299-312.

Farrell CL, Stewart PA, Del Maestro RF. A new glioma model in rat: the C6 spheroid implantation technique permeability and vascular characterization. J Neurooncol. 1987;4(4):403-15.

Giakoumettis D, Kritis A, Foroglou N. C6 cell line: the gold standard in glioma research. Hippokratia. 2018 Jul-Sep;22(3):105-112.

Gieryng A, Pszczolkowska D, Bocian K, Dabrowski M, Rajan WD, Kloss M, Mieczkowski J, Kaminska B. Immune microenvironment of experimental rat C6 gliomas resembles human glioblastomas. Sci Rep. 2017 Dec 14;7(1):17556.

Herculano-Houzel S, Mota B, Lent R. Cellular scaling rules for rodent brains.Proc Natl Acad Sci U S A. 2006 Aug 8;103(32):12138-43.

Inoue T, Fukui M, Nishio S, Kitamura K, Nagara H. Hyperosmotic blood-brain barrier disruption in brains of rats with an intracerebrally transplanted RG-C6 tumor. J Neurosurg. 1987 Feb;66(2):256-63.

Landry AP, Balas M, Alli S, Spears J, Zador Z. Distinct regional ontogeny and activation of tumor associated macrophages in human glioblastoma. Sci Rep. 2020;10(1):19542.

Noorani I, Sidlauskas K, Pellow S, Savage R, Norman JL, Chatelet DS, Fabian M, Grundy P, Ching J, Nicoll JAR, Boche D. Clinical impact of anti-inflammatory microglia and macrophage phenotypes at glioblastoma margins. Brain Commun. 2023 2;5(3):fcad176.

Petterson SA, Sørensen MD, Burton M, Thomassen M, Kruse TA, Michaelsen SR, Kristensen BW. Differential expression of checkpoint markers in the normoxic and hypoxic microenvironment of glioblastomas. Brain Pathol. 2023 Jan;33(1):e13111.

Pyonteck SM, Akkari L, Schuhmacher AJ, Bowman RL, Sevenich L, Quail DF, Olson OC, Quick ML, Huse JT, Teijeiro V, Setty M, Leslie CS, Oei Y, Pedraza A, Zhang J, Brennan CW, Sutton JC, Holland EC, Daniel D, Joyce JA. CSF-1R inhibition alters macrophage polarization and blocks glioma progression. Nat Med. 2013 19(10):1264-72.

San-Galli F, Vrignaud P, Robert J, Coindre JM, Cohadon F. Assessment of the experimental model of transplanted C6 glioblastoma in Wistar rats. J Neurooncol. 1989 Sep;7(3):299-304.

Zhang M, Sun H, Deng Y, Su M, Wei S, Wang P, Yu L, Liu J, Guo J, Wang X, Han X, He Q, Shen L. COPI-Mediated Nuclear Translocation of EGFRvIII Promotes STAT3 Phosphorylation and PKM2 Nuclear Localization. Int J Biol Sci. 2019;15(1):114-126.

---

## [Decision Letter · Decision Letter 1]

9 Jan 2024

Morphological differentiation of peritumoral brain zone microglia.

PONE-D-23-16668R1

Dear Dr. Peña,

We’re pleased to inform you that your manuscript has been judged scientifically suitable for publication and will be formally accepted for publication once it meets all outstanding technical requirements.

Kind regards,

Giuseppe Biagini, MD

Academic Editor

PLOS ONE

Additional Editor Comments (optional):

Reviewers' comments:

Reviewer's Responses to Questions

**Comments to the Author**

1. If the authors have adequately addressed your comments raised in a previous round of review and you feel that this manuscript is now acceptable for publication, you may indicate that here to bypass the “Comments to the Author” section, enter your conflict of interest statement in the “Confidential to Editor” section, and submit your "Accept" recommendation.

Reviewer #1: All comments have been addressed

Reviewer #2: All comments have been addressed

2. Is the manuscript technically sound, and do the data support the conclusions?

Reviewer #1: Yes

Reviewer #2: (No Response)

3. Has the statistical analysis been performed appropriately and rigorously? 

Reviewer #1: Yes

Reviewer #2: (No Response)

4. Have the authors made all data underlying the findings in their manuscript fully available?

Reviewer #1: Yes

Reviewer #2: (No Response)

5. Is the manuscript presented in an intelligible fashion and written in standard English?

Reviewer #1: Yes

Reviewer #2: (No Response)

6. Review Comments to the Author

Reviewer #1: (No Response)

Reviewer #2: (No Response)

7. PLOS authors have the option to publish the peer review history of their article (what does this mean?). If published, this will include your full peer review and any attached files.

Reviewer #1: No

Reviewer #2: No

---

## [Editor Report · Acceptance letter]

31 Jan 2024

PONE-D-23-16668R1 

PLOS ONE

Dear Dr. Peña-Ortega, 

I'm pleased to inform you that your manuscript has been deemed suitable for publication in PLOS ONE. Congratulations! Your manuscript is now being handed over to our production team.

Kind regards, 

on behalf of

Dr. Giuseppe Biagini 

Academic Editor

PLOS ONE